# Photosensitizer-specific bacterial stress responses in *Escherichia coli* reveal distinct targets in photoinduced inactivation

Marco Chittò[1,2], David Tutschner[1], Ulrich Dobrindt [1,2], Anzhela Galstyan [3]✉ & Michael Berger [1]✉

The ongoing antibiotic crisis calls for alternative antimicrobial strategies. Antimicrobial photodynamic therapy (aPDT) offers a sustainable option, employing a light-activated photosensitizer (PS) to generate cytotoxic reactive oxygen species (ROS). The non-selective nature of these ROS helps minimize the risk of resistance development. Despite the development of numerous PSs, limited data on their exact mechanisms and bacterial targets still hinders broader clinical use. The focus of this study is to address this gap by capturing pathway-specific responses to sub-lethal photodynamic stress using a panel of transcriptional biosensors in *E. coli* K-12 strain MG1655. Our results indicate that methylene blue (MB) primarily causes oxidative stress in the cytosol while silicon phthalocyanine derivative (SiPc) induces envelope stress at physiological conditions. By monitoring well-characterized stress response pathways, our method offers a valuable tool for elucidating the physiological effects of aPDT and guiding more detailed mechanistic or transcriptomic studies.

Antimicrobial photodynamic therapy (aPDT) has emerged as a promising and rapidly expanding therapeutic modality for the treatment of various multidrug-resistant infections[1,2]. In contrast to standard antibiotic treatment, there are several clear benefits of this approach. To the best of current knowledge, mutant bacteria resistant to aPDT have never been isolated or characterized[3], the duration of treatment is generally quite short and aPDT targets the region of interest with little to no invasiveness[4,5].

The aPDT modality employs a light-activated compound called a photosensitizer (PS) that can either be introduced topically or systemically before tissue illumination with harmless visible light. Following absorption of a photon of light of the specific wavelength according to its' set absorption spectrum, the PS is excited into a long-lived triplet state, which can react with nearby molecules by one of three proposed mechanisms. Type I mechanism results in an electron or hydrogen transfer, leading to the production of radicals. The Type II mechanism results in energy transfer directly to molecular oxygen yielding highly reactive singlet molecular oxygen, $^1O_2$[6,7] with subsequent cytotoxic effects due to the potential of reactive oxygen species (ROS) reacting with nucleic acids, proteins, and/or cell membranes, thus destroying microorganisms in a very short time frame. A third, Type III, mechanism has recently been shown where aPDT can be potentiated by the simple addition of an inorganic salt such as potassium iodide[6]. This Type III mechanism involves photoinduced electron transfer that produces reactive inorganic radicals, enabling the inactivation of bacteria in the absence of oxygen[7].

Recently, a proteomic approach has been used to characterize aPDT-induced damage to *Staphylococcus aureus*[8]. By analyzing the damaged proteins, it was found that the aPDT-induced damage to proteins is specific and likely dependent on the localization of the PS in the bacterial cell[9]. Using octacationic Zn(II)-phthalocyanine, it was demonstrated that the initial photodamage of *E. coli* cells occurs at the level of specific proteins in the outer membrane, thus promoting the penetration of the photosensitizer into the cytoplasmic membrane, where some enzymes critical for cell survival were inactivated[10].

Since the ability of some PSs to bind to bacterial membranes shows a strong correlation with their antibacterial efficacy, it has been widely accepted—though not universally proven—that the bacterial membrane serves as a primary target in aPDT[11–13]. This association has guided much of the research and development in aPDT, emphasizing the importance of membrane interactions in achieving effective bacterial inactivation[14,15]. However, other studies have reported that ROS-induced damage to DNA,

[1]Institute of Hygiene, University of Münster, Münster, Germany. [2]AO Research Institute Davos, Davos, Switzerland. [3]University of Duisburg-Essen, Faculty of Chemistry, Center for Nanointegration Duisburg Essen (CENIDE), Center for Water and Environmental Research (ZWU) and Center for Molecular Biotechnology (ZMB), Universitätsstrasse 5, Essen, Germany. ✉e-mail: anzhela.galstyan@uni-due.de; Michael.Berger@ukmuenster.de

and the resulting disturbances in DNA replication and transcription processes, kill the cells[16]. Therefore, despite many efforts, it is still not clear which of these mechanisms and which targets are most important for microbial inactivation with an aPDT, and yet the elucidation of the mechanisms and targets of aPDT remains fundamentally important, as it is the key to further improve the antimicrobial activity and utilization of PSs[17]. Unfortunately, despite all these studies, there is still a lack of experimental data on the physiological response of bacteria to various PSs. Global approaches to address this, such as proteomic or transcriptomic analysis, can be time- and cost-prohibitive and, if pursued, may result in a very limited resolution due to the reduced number of samples that may be processed. Thus, making these experimental methods unsuitable for the screening of large libraries of PSs. To address this limitation, we therefore developed high-throughput biosensors for screening the impact of PSs on microbial physiological function. The assay is based on a set of chromosomally integrated promoter-yellow fluorescent protein (yfp) gene fusion modules in E. coli K-12 MG1655 that allow monitoring the expression of representative genes of major bacterial stress response pathways. In our study on the bacterial response to the photodynamic action, we chose two different classes of PSs: MB as one of the well-studied PSs that has been extensively used for the treatment of malignant diseases and infections[18] and 2(3),9(10),16(17),23(24)-tetrakis-[3-(N-phenyl)pyridyloxy]-phthalocyaninato dihydroxy-silicon (IV) tetrabromide (SiPc)[19] as a representative of positively charged, non-aggregated tetrapyrrole-based PS (Fig. 1). Both PSs absorb light in the visible electromagnetic spectrum with the absorption maxima at 664 nm (logε = 4.89) for MB and 678 nm (logε = 5.10) for SiPc and have comparable singlet oxygen quantum yields in organic solvents ($\Phi_\Delta = 0.57$ for MB and $\Phi_\Delta = 0.59$ for SiPc). Despite very similar photophysical properties, the photodynamic effect of these PSs was found to be very different[20]. Whereas the photoactivity of MB against Gram-negative bacteria is very low, SiPc was found to decrease the bacterial viability at very low concentrations under the same irradiation conditions (Fig. S1, Supporting Information).

Our biosensors were also constructed to contain a constitutively expressed cyan fluorescent protein gene (Pfrr-cfp) for the normalization of the stress response, as the optical properties of the PSs prohibit the classical

use of normalization of reporter gene expression to the optical density of the bacteria. Using these biosensors in combination with a set of regulator mutants, we have demonstrated in this study that MB activates the oxidative stress response via OxyR, whereas SiPc induces envelope stress in the bacteria via BaeR and CpxR in a concentration- and light-dependent manner.

## Results

### Construction and testing of the biosensors

The construction of our SOS biosensor is described elsewhere[21]. In order to measure a more comprehensive bacterial response to the PSs, we constructed a set of reporter strains in an identical manner that carry transcriptional fusions of yfp to the promoters of different stress response genes. We used yfp to replace the pH stress pathway gene gadA (coding for glutamate decarboxylase A)[22–24], the envelope stress pathway gene spy (coding for spheroplast protein Y)[25–27], the osmotic stress response pathway gene otsA (coding for trehalose-6-phosphate synthase, A subunit)[28] and the oxidative stress response pathway gene dps (coding for DNA-binding protein from starved cells)[29,30]. In a second step, we PCR-amplified the stress gene promoter-yfp reporter construct from the chromosome and integrated it convergent to the constant part of the module that consisted of a transcriptional fusion of cfp to the promoter of the housekeeping gene frr (coding for ribosome-recycling factor)[31–33] in the λ attB site of the E. coli K-12 MG1655 chromosome (Fig. 2).

The Pfrr-cfp fusion construct can serve as an alternative for the normalization of stress promoter response in cases where the optical properties of test substances do not allow for OD measurements[32,34]. After Sanger sequencing of relevant module elements (chromosome-module and promoter-reporter gene junctions), we first tested the sensibility of the biosensor set containing the different modules to different concentrations of known inducers (Fig. 3).

Briefly, we diluted the reporter strains 1:200 in 150 µl of M9 medium supplemented with casamino acids and 0.4% glucose in a 96-well plate, and grew the bacteria in the microplate reader to the logarithmic growth phase, recording optical density and fluorescence signals automatically. Once the bacteria reached logarithmic growth, we added 5 µl of either a dilution series of the stress-inducing chemical or the solvent of the stress-inducing

**Fig. 1 | Photosensitizers and photochemical pathways for ROS generation.** Chemical structures of the photosensitizers methylene blue (MB) and silicon phthalocyanine (SiPc) used in this study are shown. Upon light irradiation, the triplet state of each photosensitizer generates ROS by Type I (radical formation) or Type II (singlet oxygen) pathways.

*Type I*

$$^3PS^* + S \rightarrow PS^{\cdot-} + S^{\cdot+}$$
$$^3PS^* + {}^3O_2 \rightarrow PS + O_2^{\cdot-}$$
$$PS^{\cdot-} + {}^3O_2 \rightarrow PS^{\cdot+} + O_2^{\cdot-}$$
$$O_2^{\cdot-} + O_2^{\cdot-} + H^+ \rightarrow H_2O_2 + O_2$$
$$Fe^{3+} + O_2^{\cdot-} \rightarrow Fe^{2+} + O_2$$
$$Fe^{2+} + H_2O_2 \rightarrow Fe^{3+} + OH + OH^-$$

*Type II*

$$^3PS^* + {}^3O_2 \rightarrow PS + {}^1O_2$$

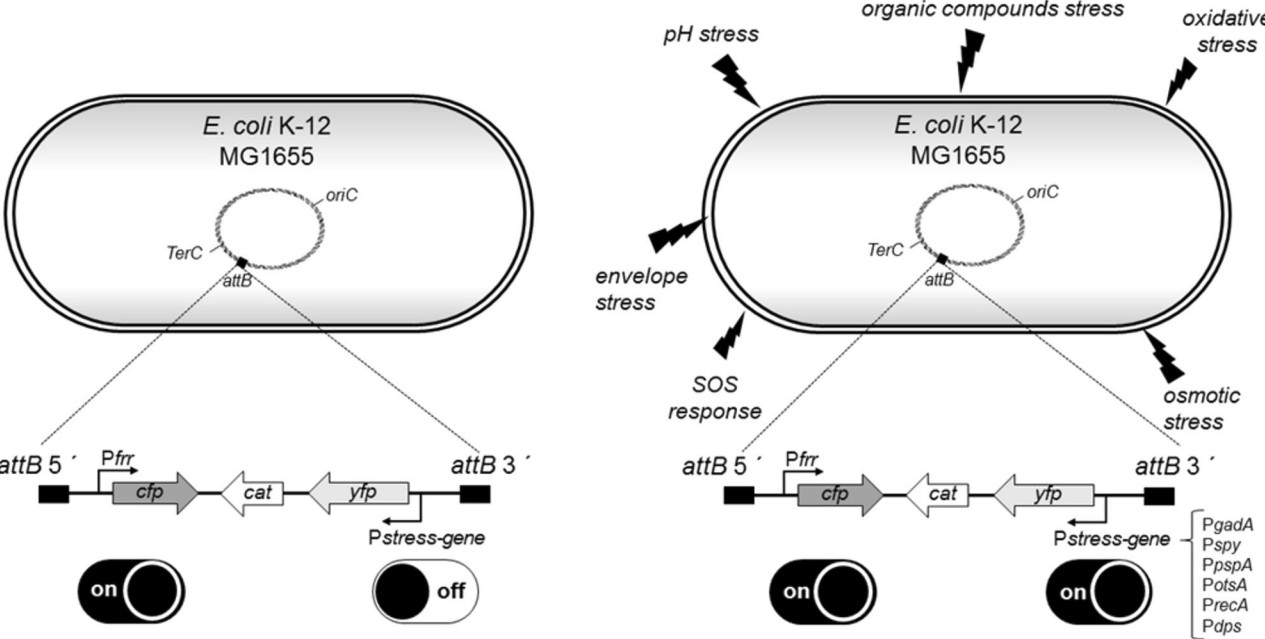

**Fig. 2 | Schematic representation of an *E. coli* K-12 MG1655 *E. coli* cell with the relevant elements.** Shown is the chromosome of, origin of replication (*oriC*), terminator of replication (*terC*) and phage λ integration site (*attB*). A graphic representation of the organization of the modules at the insertion site is indicated below.

During normal growth, the P*frr* promoter is weak, but constitutively active, while the stress promoters are switched off (left). After a challenge with a suitable stressor, the stress-related promoter becomes activated (right).

chemical (control) and continued to record the response of the bacteria. This experimental setup allowed us to simultaneously determine (i) the minimal concentration of a test substance that affects the bacterial growth rate, (ii) the final optical density of the bacterial culture in the presence of a range of concentrations of a test substance (here defined as optical density at 20 h after the start of the experiment), (iii) the expression kinetics of stress response genes towards various concentrations of a test substance and, (iv) if a stress response towards a test substance is dependent on the concentration of the test substance.

**pH stress biosensor (P*gadA* module).** We tested the pH stress biosensor (containing the *gadA* promoter module; P*gadA* module) by challenging the bacteria with acetic acid, as it was previously shown that *gadA* expression is strongly induced by acetate[21]. Adding acetic acid to 0.1% f.c. in the logarithmic growth phase (Fig. 3A, dashed line) had an impact on the growth rate of the bacteria. A mild reduction in the growth rate could still be observed at 0.05% acetic acid, but there was almost no difference at 0.025% acetic acid when compared to the control. The reduced growth rate at 0.1% acetic acid also correlated with the observed reduced final optical density at the end of the experiment (20 h), which was not the case at 0.05% acetic acid (Fig. 3B). In the absence of acetic acid stress, the YFP/CFP fluorescence signal ratio showed only a very mild increase over time from 170 to 250 min when the cells entered the stationary phase (Fig. 3C, control). This growth phase-dependent expression pattern is consistent with the previously described growth phase- and σ^S-dependent expression pattern of *gadA*[22]. In contrast to that, the YFP/CFP fluorescence signal ratio increased more strongly and in an acetic acid concentration-dependent manner after the addition of acetic acid (Fig. 3C, dashed line). This response lasted until the growth phase-dependent induction of the P*gadA* started (~250 min), and the overall YFP fluorescence signal was too strong to be recorded with the settings of the instrument around ~310 min. As the acetic acid stress response was immediately followed by the growth phase-dependent activation of P*gadA*, we calculated the total amount of YFP/CFP that was expressed for an hour following the addition of acetic acid (170 min). The total response of P*gadA* to the different concentrations of acetic acid was

concentration-dependent (Fig. 3D) and significant enough to produce marked amounts of YFP/CFP at 0.025% acetic acid, which did not have a noticeable effect on growth rate or final optical density (Fig. 3A, B). We did not observe any concentration-dependent response of the other biosensors to the addition of acetic acid (Fig. S2, black arrows).

**Envelope stress biosensor (P*spy* module).** We tested the envelope stress biosensor (containing the *spy* promoter module; P*spy* module) by challenging the bacteria with tannic acid, which was described as a strong inducer of *spy* expression[35]. As shown in Fig. 3E, the addition of up to 67 µg/ml tannic acid (*t* = 170 min, dashed line) had only a mild impact on the growth rate and resulted in only a slight, concentration-dependent reduction of the final optical cell density at the end of the experiment (Fig. 3F). This finding was consistent with previous studies that showed growth inhibition at higher tannic acid concentrations[36]. In our study, the YFP/CFP fluorescence signal ratio started to increase immediately after the addition of tannic acid, and the increase was concentration-dependent, lasting approximately to time point 270 min, after which no increase in the fluorescence signal was observed (Fig. 3G). In the absence of tannic acid, the P*spy* module only expressed very small amounts of YFP/CFP over the whole course of the experiment, which was consistent with previous studies[35]. As the response to tannic acid effectively ended at ~270 min, we used this endpoint for our calculations, calculating the total increase in YFP/CFP from the time point of induction until 270 min (Fig. 3H). We did not observe a concentration-dependent response of the other biosensors following the addition of tannic acid (Fig. S3, black arrows).

**Osmotic stress biosensor (P*otsA* module).** We tested the osmotic stress biosensor (containing the *otsA* promoter module; P*otsA* module) by challenging the bacteria with different concentrations of NaCl (Fig. 3I–L). The challenge was relatively mild and affected the growth rate only at concentrations of 500 mM NaCl (Fig. 3I). However, the final OD at the end of the experiment was not affected (Fig. 3J). Similar to the P*gadA* module, the P*otsA* module started expressing YFP in the absence of an external stimulus at the transition to the stationary growth phase, as

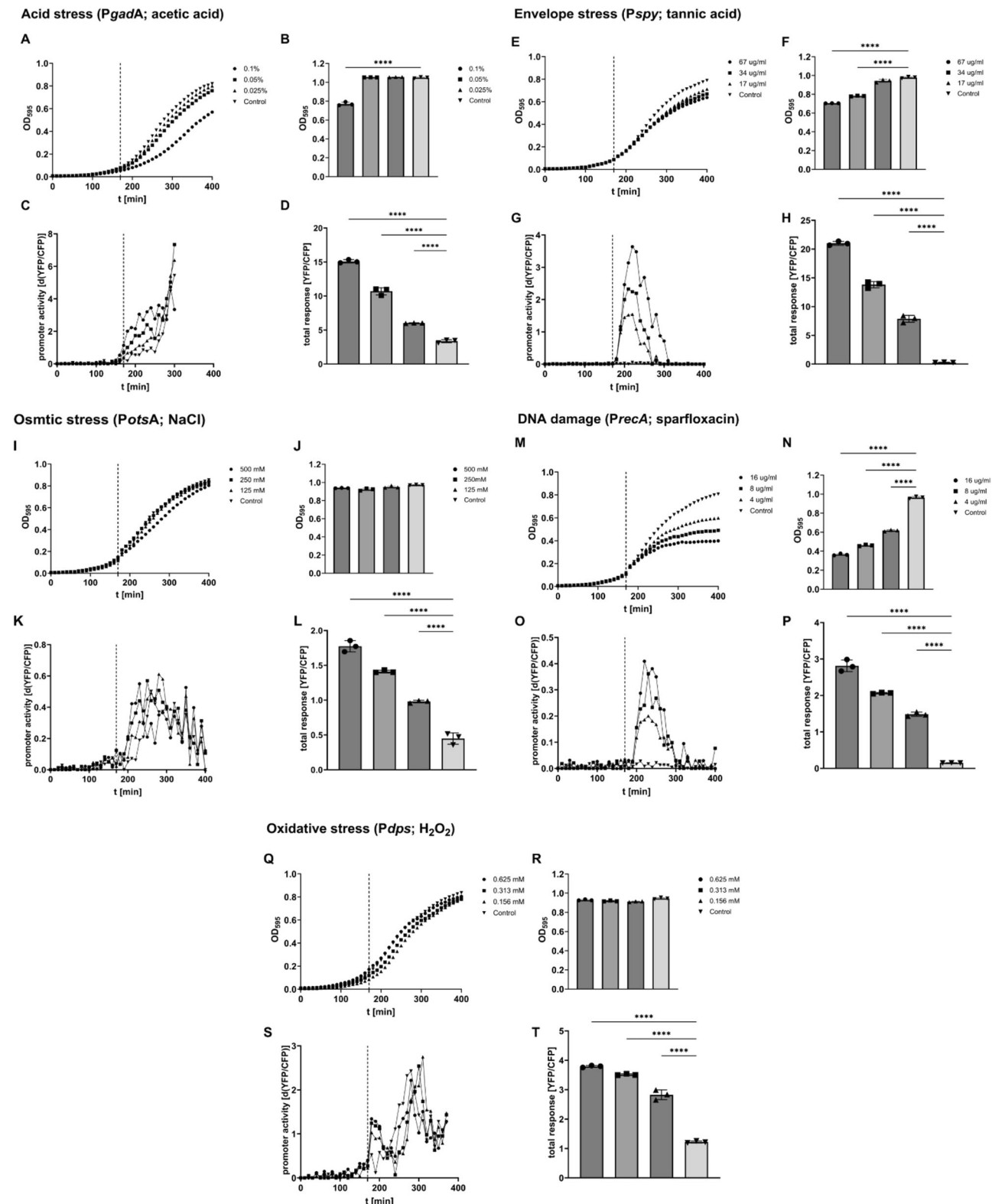

**Fig. 3 | Test of the reporter strains with known stressors.** Shown are the relevant parts of the growth curves (**A, E, I, M, Q**), final optical densities at the end of the experiments (20 h; **B, F, J, N, R**), the YFP/CFP ratio increase of the indicated module in response to a stressor over time (**C, G, K, O, S**) and the total YFP/CFP expression in response to a stressor (**D, H, L, P, T**). The stressor and the tested modules are indicated on top of the graphs, and the tested concentrations and corresponding symbols are on top of the growth curves. The time point of the addition of the stressor is indicated with a dashed line in growth curves and promoter kinetics. For the details, see the text.

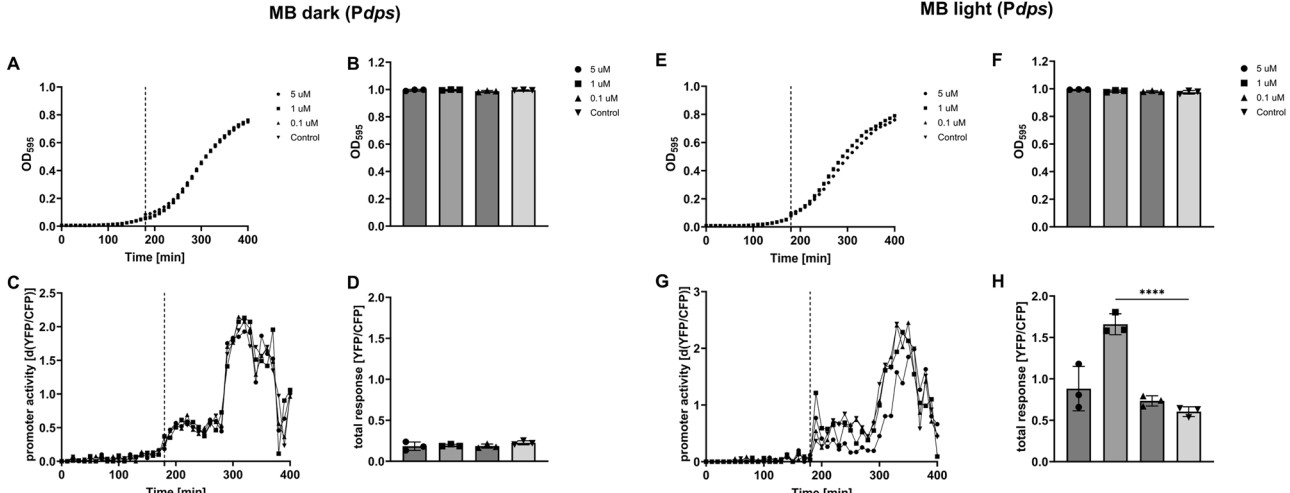

**Fig. 4 | MB is inducing oxidative stress in *E. coli* K-12 MG1655 after light activation.** *MB without photo activation (MB dark).* Shown are growth curves of the reporter strain (**A**), final optical densities (**B**), YFP/CFP expression of the P*dps* module over time (**C**), and total YFP/CFP expression (**D**) in response to the indicated concentration of MB. *MB with photo activation (MB light).* Shown are growth curves of the reporter strain (**E**), final optical densities (**F**), YFP/CFP expression of the P*dps* module over time (**G**), and total YFP expression (**H**) in response to the indicated concentration of MB after photo activation. The dashed lines indicate the time point of the addition of the test substance. Shown are the results of representative experiments. The columns (**D**, **H**) represent the average results of three biological replicates (*$P < 0.05$; **$P < 0.01$; ***$P < 0.001$).

judged by the continuously increasing YFP/CFP ratio after ~230 min (compare control in Fig. 3K with control in Fig. 3C). The addition of NaCl to a final concentration of 500 mM, 250 mM, and 125 mM NaCl resulted in a concentration-dependent response of the P*otsA* module (Fig. 3K). As the osmotic stress response was followed by the growth phase-dependent activation of P*otsA*, we calculated the concentration-dependent response for the P*otsA* module for 1 h after the NaCl challenge (Fig. 3L). None of the other biosensors showed a concentration-dependent response to NaCl (Fig. S4).

**SOS response biosensor (P*recA* module).** We have previously tested the functionality of the SOS response biosensor module (containing the *recA* promoter module; P*recA* module) by the assessment of an enzymatically inhibited SOS response[21]. Here, we tested the P*recA* module by exposing the bacteria to different concentrations of the gyrase inhibitor sparfloxacin[37] (Fig. 3M–P, dashed lines). Sparfloxacin concentration-dependent effects on the growth rate were observable from approximately 200 min (Fig. 3M), and the final optical density was sparfloxacin concentration-dependent as well, as expected for an antibiotic (Fig. 3N). The response of the P*recA* module to the sparfloxacin challenge at 170 min (dashed line) was rapid and lasted until ~300 min (Fig. 3O). In this period, the P*recA* module of the control sample expressed only small amounts of YFP/CFP, whereas the total amount of YFP/CFP expressed in response to sparfloxacin was dependent on the concentration (Fig. 3P). None of the other biosensors was induced by the addition of sparfloxacin (Fig. S5). In contrast, the growth phase-dependent activation of the P*gadA*, P*otsA,* and P*dps* modules showed a sparfloxacin-dependent reduction in YFP/CFP produced per period, as expected.

**Oxidative stress biosensor (P*dps* module).** We have previously tested the functionality of the oxidative stress biosensor (containing the *dps* promoter module; P*dps* module) in *E. coli* K-12 extensively[30]. The different concentrations of $H_2O_2$ tested here did not affect the growth rate (Fig. 3Q) and the final cell density at the end of the experiment (Fig. 3R). The P*dps* module started to express YFP rapidly after the challenge with $H_2O_2$ as judged by the rapid increase in YFP/CFP (Fig. 3S) and the response lasted for ~30 min under these conditions, similar to what we have observed earlier[30]. The growth phase-dependent induction of the P*dps* module in the absence of $H_2O_2$ stress started around 220 min in this

experiment, whereas the growth phase-dependent activation after the challenge with $H_2O_2$ was delayed (Fig. 3S). In order to separate the $H_2O_2$ response from the growth phase-dependent activation of the P*dps* module, we calculated the total amount of YFP/CFP expressed for 30 min after induction. As for the other chemicals tested, the total response depended on the concentration of the test substance (Fig. 3T). The P*gadA* and P*otsA* modules were not induced by $H_2O_2$, but showed a delay in the growth phase-dependent activation, similar to the P*dps* module (compare Fig. S6 to Fig. 3S). The $H_2O_2$ challenge did not induce the SOS biosensor as strongly as sparfloxacin (compare Fig. S6 to Fig. 3O). However, the YFP/CFP ratio was continuously increasing after the addition of $H_2O_2$ from 180 min to ~260 min, indicative of an induced bacterial subpopulation (see Fig. S11 of Supporting Information).

**Photosensitizers induce different types of stress in *E. coli***
**Oxidative stress induced by light-activated MB.** Next, we tested different concentrations of MB for the activation of our biosensor set. In order to better separate a potential acid, osmotic, or oxidative stress response from the growth phase-dependent activation of the promoter, we added the test substance earlier (140 min; Fig. 4, dashed line). Without photoactivation, no effects of the tested concentrations of MB on the growth rate (Fig. 4A), final optical cell density (Fig. 4B), or induction of the oxidative stress biosensor were observed (Fig. 4 C). In addition, the growth phase-dependent activation of the module was not affected at all MB concentrations tested (Fig. 4C). MB also did not induce the rest of the biosensors without photoactivation at all concentrations tested (Fig. S7). In contrast, the P*dps* module started to express YFP immediately after photoactivation of MB at all concentrations tested (note the YFP/CFP peak at 150 min in Fig. 4G). The final amount of YFP/CFP that was expressed in response to the photoactivation was dependent on the concentration of MB as well, however, higher at 1 μM MB than at 5 μM MB (Fig. 4H). There was only a mild effect on the growth rate of the bacteria observable after photoactivation of 5 μM MB (Fig. 4E), but no effect on final cell density (Fig. 4F) at all concentrations tested. We also tested the P*dps* module on a plasmid[38] in an *E. coli* K-12 MG1655 *rpoS* mutant and observed induction after photoactivation of MB as well, which showed that MB is indeed activating the P*dps* module via the σ[70] holoenzyme and not due to a general RpoS-mediated stress response (Fig. S8). None of the other biosensors showed induction after light

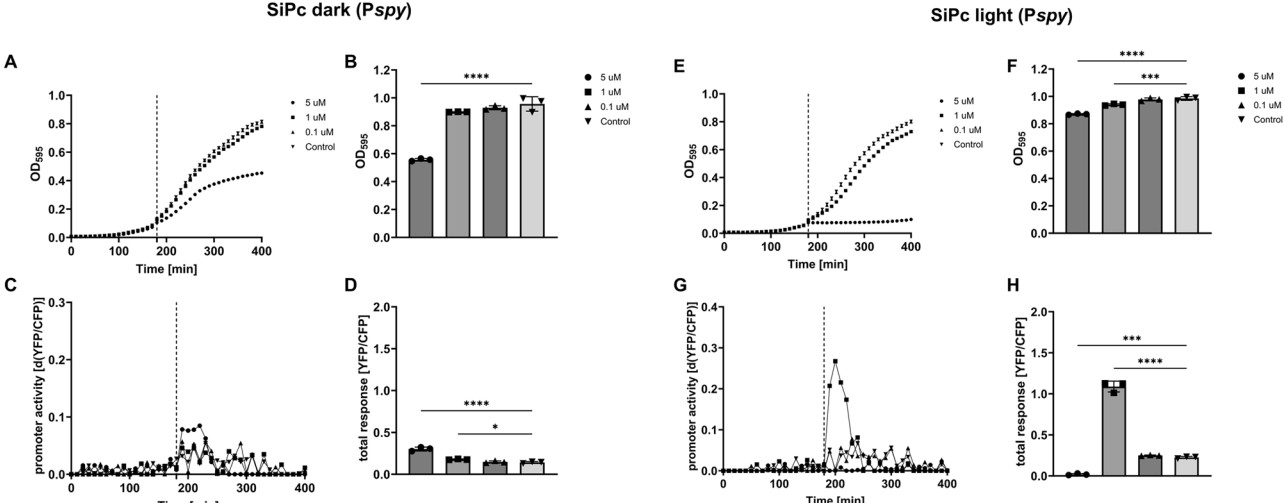

**Fig. 5 | SiPc is inducing envelope stress in *E. coli* K-12 MG1655.** *SiPc without photo activation (SiPc, dark).* Shown are growth curves of the reporter strain (**A**), final optical densities (**B**), YFP/CFP expression of the P*spy* module over time (**C**), and total YFP expression (**D**) in response to the indicated concentration of SiPc. At 5 μM, the P*spy* module shows a mild response (**C**) that is associated with reduced growth rate (**A**) and a reduced final optical density at the end of the experiment (**B**). *SiPc with photo activation (SiPc, light).* Shown are growth curves of the reporter strain (**E**), final optical densities (**F**), YFP/CFP expression of the P*spy* module over time (**G**), and total YFP/CFP expression (**H**) in response to the indicated concentration of SiPc

after photo activation. Light activation of SiPc results in an immediate stop in the increase of OD at 5 μM SiPc (**E**), and the response of the P*spy* module to 1 μM SiPc is stronger than to 0.1 and to 5 μM SiPc, at which the response is actually reduced when compared to the control (**G**, **H**). The dashed lines indicate the time point of the addition of the test substance. Shown are the results of representative experiments. The columns (**D**, **H**) represent the average results of three biological replicates (*$P < 0.05$; **$P < 0.01$; ***$P < 0.001$).

activation of MB at all concentrations tested (Fig. S9). However, the P*gadA* and P*otsA* modules showed MB concentration-dependent time delay in growth phase-dependent activation, similar to what we have observed for $H_2O_2$ (compare P*gadA* and P*otsA* modules in Fig. S9 to Fig. S6).

**Envelope stress induced by light-activated SiPc.** We next tested different concentrations of SiPc for the activation of our biosensor set. At all concentrations tested, SiPc did not activate the biosensors without photoactivation (Fig. S10), except for envelope stress. The P*spy* module that showed a mild induction at 5 μM SiPc (Fig. 5C), resulted in slightly elevated total YFP/CFP expression as well (Fig. 5D). In addition, the growth rate was affected without photo activation at 5 μM SiPc (Fig. 5A), which also resulted in a reduced optical density at the end of the experiment (Fig. 5B). The other biosensors did not show any induction in the absence of photo activation, but the growth phase activation of the P*gadA*, P*otsA* and P*dps* modules was either slightly delayed at 1 μM SiPc, or completely absent at 5 μM SiPc (Fig. S10). Photoactivation at a concentration of 5 μM SiPc caused an immediate stop of bacterial cell growth and/or cell division as judged by the stable optical density upon exposure to red light, whereas photoactivation at 1 μM SiPc had only a mild effect on the growth rate (Fig. 5E). However, in spite of the immediate and long-lasting effect on the growth of the bacteria (resuming growth was detectable at ~400 min), photoactivation of 5 μM SiPc reproducibly resulted in higher final optical densities at the end of the experiment than 5 μM SiPc without photo activation (compare Fig. 5B and 5F). Photoactivation resulted in an immediate induction of the P*spy* module that lasted until ~200 min only at 1 μM SiPc, but not at the other concentrations tested (Fig. 5G). A significant total YFP/CFP expression in this period was also only observed for 1 μM SiPc (Fig. 5H). None of the other biosensors was induced after light activation of SiPc. In contrast, the growth phase-dependent activation of the P*gadA*, P*otsA,* and P*dps* modules was either delayed or weaker at 1 μM SiPc, or completely absent at 5 μM SiPc in this period (Fig. S11).

This is important, as higher concentrations of substances that kill bacteria are also very likely to interfere with the expression of reporter genes, as seen here at 5 μM MB and 5 μM SiPc (Figs. 4G and 5G).

## Light-activated oxidative stress induced by MB is sensed by OxyR, envelope stress induced by SiPc is sensed by BaeR and CpxR

The fact that, except for the P*dps* and P*spy* modules, none of the other biosensors were induced following photoactivation of MB and SiPc indicated that the *dps* and *spy* promoters were not induced due to a general stress response. That the P*dps* module was also induced by light activation of MB in an *rpoS* mutant background further supported the hypothesis that the P*dps* module was specifically activated. However, if the activation of both modules by the stressors was specific, it should be absent in the absence of the relevant response regulators. In order to test this hypothesis, we decided to delete the regulators of P*dps* and P*spy* and in the oxidative stress and the envelope stress biosensor, respectively. Afterwards, we compared the response of the mutant biosensors to the PSs to the wild type.

**Photoactivation of MB induces the P*dps* module via OxyR.** The *dps* promoter is regulated via OxyR and MntR[39,40]. We deleted both genes in our oxidative stress biosensor strain by recombineering and repeated the experiment with photoactivation of 1 μM MB (Fig. 6). In the mntR-negative strain background, the response of the Pdps module to light activation of 1 μM MB was very similar to wild type and showed a prominent peak (Fig. 6A, 150 min) in Pdps module activity directly following the treatment (Fig. 6A, dashed line). This peak in promoter activity was completely absent in the control experiment (Fig. 6B). Notably, this peak in Pdps activity following the treatment was also completely absent in the oxyR mutant (Fig. 6C), and the module showed a very similar response in comparison to the water control experiment in the oxyR mutant (Fig. 6D). The total response of the modules to photoactivation of 1 μM MB was also similar in wild type and *mntR* mutant, but reduced to control levels in the *oxyR* mutant (compare Fig. 6E with Fig. 6F).

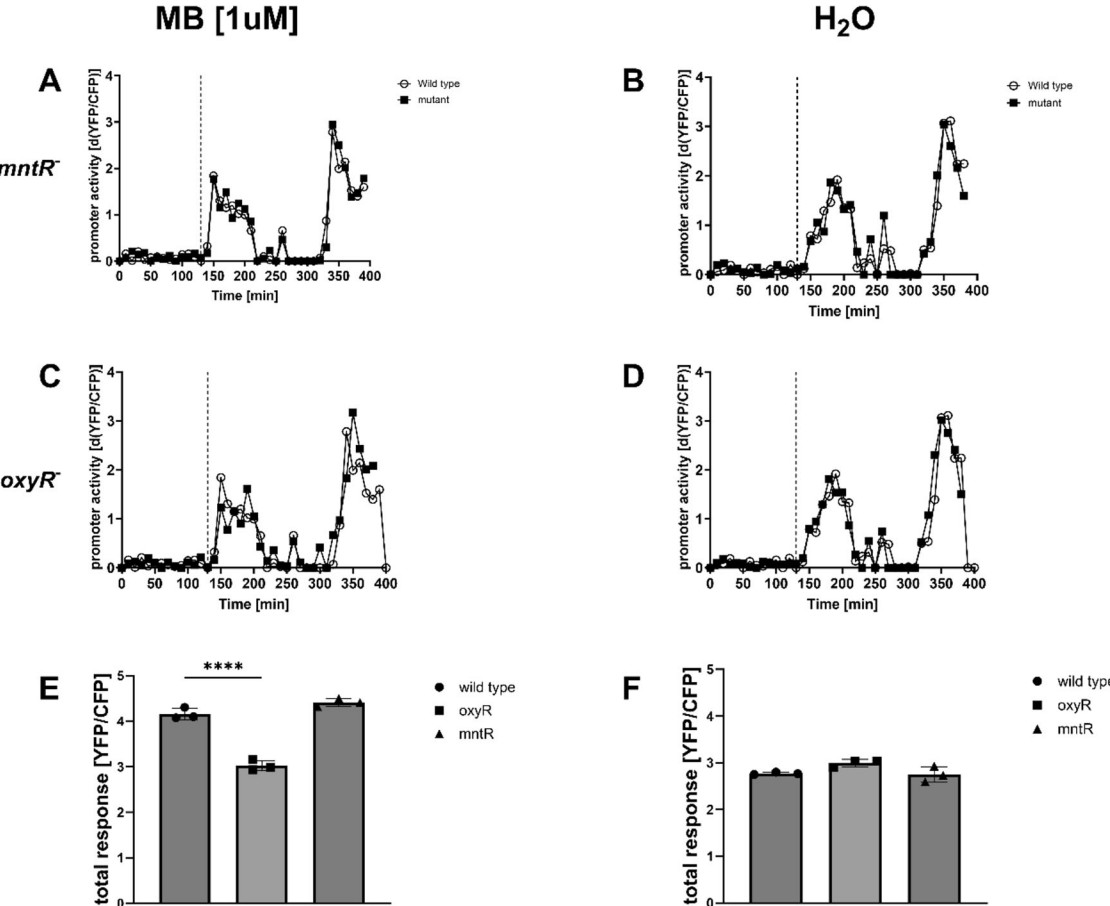

**Fig. 6 | The MB-dependent induction of P*dps* is not affected in mnt*R* mutants, but absent in *oxyR* mutants.** Shown is the YFP/CFP expression in response to MB after photoactivation in wild-type and *mntR* mutant (**A**) and wild-type and *oxyR* mutant (**C**), as well as the respective controls (**B**, **D**). As opposed to *mntR* mutants, the total response of the Pdps module to MB after light activation is significantly reduced in *oxyR* mutants (**E**) and very similar to the water control (**F**). The dashed lines indicate the time point of the addition of the test substance. Shown are the results of representative experiments. The columns (**E**, **F**) represent the average results of three biological replicates (*$P < 0.05$; **$P < 0.01$; ***$P < 0.001$).

**Photoactivation of SiPc induces the P*spy* module via BaeR and CpxR.** The *spy* promoter is regulated by the two-component system BaeSR and the regulator CpxR[41]. We deleted *baeR* and *cpxR* in our envelope stress biosensor strain by recombineering and repeated the experiment with photoactivation of 1 μM SiPc (Fig. 7). In the *cpxR* mutant (Fig. 7A), as well as in the *baeR* mutant (Fig. 7B) the P*spy* promoter module response following photoactivation of SiPc was strongly reduced when compared to the wild-type. The total response of the P*spy* module was reduced in the *cpxR* mutant relative to the wild-type (Fig. 7E) and not detectable in the *baeR* mutant, where the P*spy* module response was indistinguishable from the water control (compare Fig. 7E, F).

## Discussion

Due to the current antibiotic crisis, aPDT has emerged as an alternative approach for antimicrobial therapy. However, the knowledge of the primary site of attack of photosensitizers is still limited[42]. Classical global approaches (e.g., proteomics or transcriptomics) used to identify sites of attack of different photosensitizers still have limitations with respect to the number of samples that can be processed. In addition, these approaches offer a very limited temporal resolution of the physiological response of the bacteria to the challenge. We therefore decided to analyze the response of a limited set of well-characterized stress gene promoter-reporter gene modules to a range of concentrations of PSs at a high temporal resolution. In order to avoid the limitations associated with the use of plasmids (such as the requirement for selection or stress-dependent copy number modifications)[43], we integrated our stress reporter modules in the λ *attB* site of the chromosome of the well-

characterized *E. coli* K-12 strain MG1655 (Fig. 2). These modules contained a construct element that provides constitutive expression of *cfp* from the promoter of the housekeeping gene *frr* so that there is a normalization alternative available if the optical properties of a test substance do not permit optical density measurements[31–33]. The second part of the modules was a construct that expresses YFP under control of a set of different stress-inducible promoters[21–27].

Our initial tests with substances that were described to activate the stress gene promoters showed that all biosensors used in this study exhibited a concentration-dependent effect to the challenges, albeit to varying degrees (Fig. 3). The absolute amount of YFP/CFP expressed in response to a challenge was dependent on the reporter module. For example, the maximal total amount of YFP/CFP expressed by the P*spy* module was ~20 YFP/CFP at the highest tannic acid concentration (Fig. 3H), whereas the maximal total amount of YFP/CFP in response to NaCl was 1.8 YFP/CFP for the P*otsA* module (Fig. 3L). This result was expected, as the strength and extent of the response of the promoter is adjusted to the specific features of the corresponding stress response gene. These differences in fluorophore expression made it difficult to record the expression of the reporter gene for all stress reporter modules over the full time period of the experiment, as can be observed with the strong response of the P*gadA* module (Fig. 3C). Nevertheless, the detection of the stress response in logarithmic growth phase was readily possible for all biosensors used in this study with identical instrument settings (Fig. 3). In the future, N-terminal modifications of the coding sequence of the reporter gene could be used to further fine-tune the fluorophore expression of the modules to each other on the translational level[44].

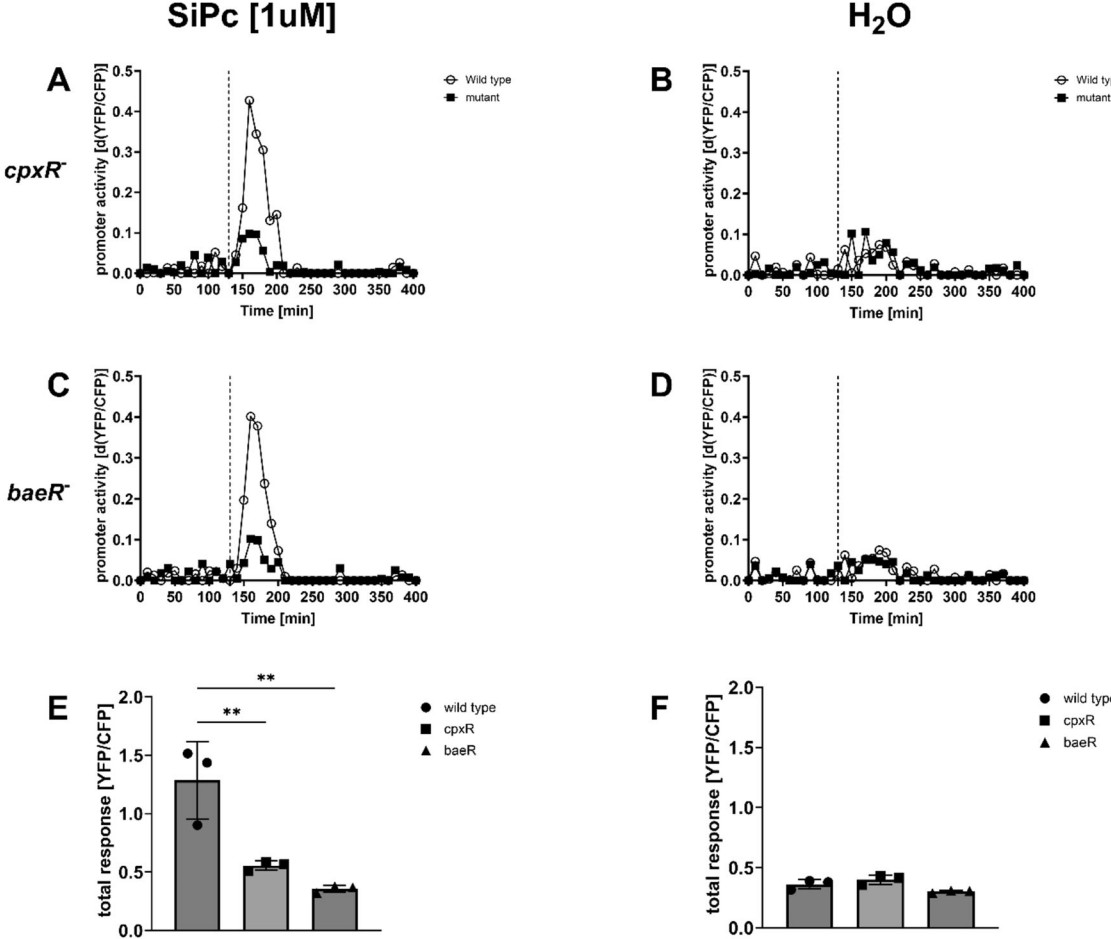

**Fig. 7 | The SiPc-dependent induction of P*spy* is reduced in *cpxR* mutants and not detectable in *baeR* mutants.** Shown is the YFP/CFP expression in response to SiPc after photoactivation in wild-type and *cpxR* mutant (**A**) and wild-type and *baeR* mutant (**C**), as well as the respective controls (**B**, **D**). The response of the P*spy* module to SiPc after light activation is reduced in *cpxR* mutants (**E**) and completely absent in baeR mutants when compared to wild type (compare **E** and **F**). The dashed lines indicate the time point of the addition of the test substance. Shown are the results of representative experiments. The columns (**E**, **F**) represent the average results of three biological replicates (*$P < 0.05$; **$P < 0.01$; ***$P < 0.001$).

Another problem was the growth phase-dependent expression patterns of individual modules (see, for example, the activity of the control sample for P*gadA* and P*dps* in Fig. 3C, S). One possibility to address this problem is to adjust the time point of the challenge in a way that allows for a better separation of the stress response from the growth phase-dependent activation of the module, which we have also done here (Figs. 3–6). Another possibility is the use of *rpoS* mutants, as long as the stress response of a module is not affected by the absence of RpoS. One such example is the P*dps* module, which does not require RpoS for the OxyR-dependent activation in response to oxidative stress in the logarithmic growth phase (Fig. S7)[39]. However, this will not be possible for the modules that are strictly dependent on RpoS, e.g., the P*gadA* and P*otsA* modules. Notably, these modules were reacting relatively slowly in response to known stressors (compare the rapid YFP/CFP increase of P*dps* and P*spy* module in Fig. 3S, G with the time-delayed increase in YFP/CFP for the P*gadA* and P*otsA* modules in Fig. 3C, K, respectively). Such a promoter kinetic would be consistent with an induction mechanism that first requires certain levels of expression and/or functional activation of an upstream regulator, here in particular RpoS that also results in a stress-independent, growth phase-dependent activation of the genes in the *rpoS* regulon[45]. Another problem was the "cross-sensitivity" of *E. coli* stress gene promoters to a variety of stressors, especially if the promoter of a stress response gene is also recognized by RpoS. In such cases, a more comprehensive analysis of the response kinetics of the modules might help to dissect direct and indirect inducing effects of a stressor on a promoter, e.g., in the delayed responses of the P*gadA* and P*otsA* modules (see above).

Notably, the purpose of the functional tests here was merely to check, if we could measure a promoter response in high throughput in a 96-well plate format, e.g., an YFP/CFP signal significantly different from the control sample following the addition of a described test substance. Identically constructed reporter modules, e.g., for the stress genes *relA*, *sulA,* and *dinB*, did not show a measurable response to the described inducers. As these modules were therefore not suitable for us to detect the stressor within the framework of our experimental settings, we simply excluded them from our analysis in this instance. The use of 96-well plates for the assays usually allowed here for the rapid identification of a suitable concentration of a test substance for the read-out of a physiological response of the reporter modules without preliminary experiments. Testing a substance at a concentration that induces a physiological response of the bacteria, such as concentrations that do not cause a drop in OD, or an immediate and persistent growth arrest, was possible with just one test run. In order to confirm that a stress response following the addition of a test substance was indeed specific for the stress caused by the substance, it was necessary to measure the responses of several independent reporter modules and compare these to the responses of known stressors (compare e.g., Fig. 3 to Figs. S1–S5)[22–30]. Whereas the specific activation of a module was already the first evidence that a certain substance was indeed inducing a specific type of stress in the bacteria, the absence of a promoter response in mutants of

regulators could provide additional evidence that a promoter was indeed specifically activated/de-repressed in response to a stressor[25–27,29,30]. Using a set of mutants of the regulators of P*spy*[41] and P*dps*[39,40] modules, we showed here that SiPc activated P*spy* via CpxR and BaeR, whereas MB induced P*dps* via OxyR (Figs. 6 and 7). Future experiments will have to show whether other stress pathways that were not investigated here are also activated by SiPc or MB. Nevertheless, the parameters of the response of P*dps* to MB and the response of P*spy* to SiPc that were identified here are very likely to be helpful to effectively choose chemical concentrations and sampling time points for more comprehensive analyses of the global bacterial responses to MB and SiPc in aPDT by other methods, such as RNA-seq.

In aPDT, oxidative stress is believed to be a major factor that affects the viability of bacteria, and one of the protective cellular mechanisms is the DNA repair system. Remarkably different responses of the oxidative stress biosensor were detected for the two PSs used. A significant and dose-dependent increase in YFP production was observed with MB upon irradiation, whereas *E. coli* treated with SiPc did not activate the P*dps* module. This could be related to the different mechanisms of action of the PSs. In diluted solutions, where MB is in a monomeric state, it acts as a type II photosensitizer generating a high amount of $^1O_2$ similar to most of the tetrapyrrole-based PSs, including SiPc[7]. However, pH[46] and different components of the biological media[20] can induce the formation of dimers and higher-order aggregates, where electron transfer reactions occur. This shifts the photosensitization mechanism to type I, promoting the generation of the superoxide anion radical $O_2^{\cdot-}$, which could enzymatically or through protonation be converted into other types of ROS, including highly cytotoxic hydrogen peroxide or hydroxyl radicals. Incubation of *E. coli* K-12 MG1655 with 5 µM of SiPc and irradiation resulted in an immediate stop of the increase of $OD_{595}$, whereas 1 µM SiPc had no marked effect on the growth rate (Fig. 5E). Nevertheless, the envelope stress biosensor was only induced at 1 µM SiPc, but not at higher, or lower concentrations of SiPc (Fig. 5G). This indicated that photoactivated SiPc, unlike MB, induced damage to the cell membrane, and that the concentration range in which a physiological envelope stress response of the bacteria to SiPc was both required and possible was relatively narrow. Interestingly, we observed a reduction of the growth rate at 5 µM SiPc in the absence of irradiation (Fig. 5A), which, in contrast to the irradiated samples, also resulted in a marked reduction of the final optical density at the end of the experiment (compare Fig. 5B to Fig. 5F). Under these conditions, the P*spy* module showed activation after 5 µM SiPc treatment, indicating that SiPc was also, in the absence of light activation, to some extent affecting the normal membrane functions. This overall indicated that unlike MB, photoactivated SiPc mainly induced damage to the cell membrane. However, this was not surprising, because as a result of a very short lifetime of $^1O_2$, aPDT-induced damage likely occurs only on the localization site of SiPc at concentrations that still allow for a physiological response of the cells[47]. Indeed, the available literature strongly supports the hypothesis that hydrophobic tetrapyrrole-based PSs tend to accumulate in the cell membrane[48–50]. Light activation of MB on the other hand is known to be mutagenic in *E. coli* since a long time and was shown to result in lesions of chromosomal DNA[51,52]. The authors also reported no activation of the SOS system following aPDT with MB, which may be surprising, however, consistent with our data on the population level (Fig. S8). In order to damage DNA by $^1O_2$ generation light activation of MB needs to occur in close spatial proximity. MB was shown to be a substrate for bacterial multi-drug efflux pumps, and the attachment of a multi-drug efflux pump inhibitor to MB was shown to enhance the efficacy of aPDT both in vitro and in vivo[53,54]. Therefore, it is conceivable that due to its small size and hydrophilic characteristic, light activation of MB in contrast to SiPc indeed results in a substantial generation of $^1O_2$ in the cytosol, which in turn results in an oxidative stress response and increase in the number of cells with activated SOS response within the bacterial population.

We have now identified both, the primary target compartment, as well as the reaction conditions at which PS-dependent modifications of components of the primary target compartment should be readily detectable. The latter is important, as reaction conditions that do not allow a

physiological response of the bacteria are also likely to result in the disintegration of the cell, which could in turn permit the PS-dependent modification of cellular components that are not the primary target of the PSs. Therefore, an analysis of marker molecules of the primary target compartments should finally clarify the question, if SiPc-mediated damage is indeed mainly occurring at the membrane, whereas MB-mediated damage is rather found in the cytosol. In addition, our analysis revealed that SiPc was both inducing envelope stress in the absence of light activation as well as after light activation. This may be a basis for a potential resistance development to PSs and should eventually be taken into consideration in therapeutic regimes in the future as well.

## Conclusion

This study demonstrates that chromosomally integrated stress reporter modules enable high-resolution, high-throughput assessment of bacterial stress responses to PSs used in aPDT. Using a panel of biosensors, we identified distinct stress signatures induced by MB and SiPc, suggesting different primary targets and mechanisms of action. MB primarily induced oxidative stress responses, consistent with cytosolic $^1O_2$ generation, while SiPc mainly caused envelope stress, indicating membrane-associated damage. These insights provide a foundation for targeted investigations into PS mechanisms and support the development of optimized aPDT strategies with minimized resistance potential.

## Methods
### Bacterial strains used in this study
All bacterial strains used in this study are listed in Table S1.

### Media and growth conditions
For the construction procedures, bacteria were grown in lysogeny broth (LB; containing 10 g/L tryptone, 5 g/L yeast extract, 5 g/L NaCl) at 37 °C under aerobic shaking conditions. For solid medium, agar was added to a final concentration of 1.5% (w/v). If necessary, the medium was supplemented with kanamycin [30 µg/ml], ampicillin [100 µg/ml], and chloramphenicol [25 µg/ml]. For the stress response measurements, bacteria were grown in M9 minimal medium (M9 salts containing 33.9 g/L $Na_2HPO_4$, 15 g/L $KH_2PO_4$, 5 g/L $NH_4Cl$, 2.5 g/L NaCl) supplemented with 0.4% casamino acid and 0.4% glucose.

### Oligonucleotides
All oligonucleotides used in this study are listed in Table S2.

### Plasmids
All plasmids used in this study are listed in Table S3.

### Construction of bacterial strains
**Construction of *E. coli* K-12 MG1655 P*stress* response gene-*yfp-cat*.** For the construction of the stress gene promoter *yfp* fusions of *gadA*, *spy* and *otsAB* the pMB54[30] vector was used as template to generate PCR products in combination with the Q5® High-Fidelity DNA polymerase and primers, MC162/MC163, MC166/MC167 and MC178/MC179 and Q5® High-Fidelity DNA polymerase, respectively. The chromosomal fusions to the respective promoters were done in *E. coli* K-12 strain MG1655 by RedE/T recombineering[55]. The verification of the correctness of the stress gene promoter-reporter gene fusion was afterwards done by amplifying the 5′ and 3′ junctions of the reporter and the chromosome by PCR. The 5′ junction PCR products (stress gene promoter – *yfp*) were additionally Sanger sequenced.

**Construction of *E. coli* K-12 MG1655 *att*::P*frr-cfp-cat-yfp*-P*stress* response gene-*yfp*.** For the construction of the final reporter modules, *E. coli* K-12 MG1655 P*stress response gene-yfp-cat* were used as templates for PCR with primers MC164/MC206, MC168/MC207, MC180/MC209 and Q5® and High-Fidelity DNA polymerase. In addition, pMB54[30] was used as a template for PCR with the primers MBPD155/MC185 Q5® and

High-Fidelity DNA polymerase. These intermediate templates were used in combination with the Q5® High-Fidelity DNA polymerase and the primers MC187/MC191, MC187/MC194, MC187/MC197, and MC187/MC183, respectively to generate DNA fragments for the final recombination step. The PCR products were then integrated convergent to the constant part of the module into the chromosome of *E. coli* K-12 MG1655 *att*::P*frr-cfp-aph(3')-Ia* with RedE/T recombineering[55]. The integration process resulted in the precise replacement of the *aph(3')-Ia* gene with the P*stress response gene-yfp-cat* fragments. The construction was completed by Sanger sequencing of the 5´and 3´ junctions amplified with the primers MBP206/MBP279.

#### Construction of *E. coli* K-12 MG1655 *att*::P*frr-cfp-cat-yfp*-P*dps mntR::aph*. *E. coli* K-12 MG1655 *att*::P*frr-cfp-cat-yfp*-P*dps mntR::aph* was constructed by transforming pKD46 into *E. coli* K-12 MG1655 *att*::P*frr-cfp-cat-yfp*-P*dps*. Afterward, we generated a PCR product using the primers MBP376/MBP377 and pKD4 as a template with the Q5® High-Fidelity DNA polymerase and replaced *mntR* with *aph* by recombineering[55]. The 5´and 3´ junctions of chromosome and *aph* were afterwards verified by PCR with primers MBP378/UD2822 and MBP373/MBP379, respectively.

#### Construction of *E. coli* K-12 MG1655 *att*::P*frr-cfp-cat-yfp*-P*dps oxyR::aph*. *E. coli* K-12 MG1655 *att*::P*frr-cfp-cat-yfp*-P*dps oxyR::aph* was constructed by transforming pKD46 into *E. coli* K-12 MG1655 *att*::P*frr-cfp-cat-yfp*-P*dps*. Afterward, we generated a PCR product using primers MBP371 / MBP372 and pKD4 as template with Q5® High-Fidelity DNA polymerase and replaced *oxyR* by *aph* by recombineering[55]. The 5´ and 3´ junctions of chromosome and *aph* were afterwards verified by PCR with primers MBP374/UD2822 and MBP373/MBP375, respectively.

#### Construction of *E. coli* K-12 MG1655 *att*::P*frr-cfp-cat-yfp*-P*spy baeR::aph*. *E. coli* K-12 MG1655 *att*::P*frr-cfp-cat-yfp*-P*spy baeR::aph* was constructed by transforming pKD46 into *E. coli* K-12 MG1655 *att*::P*frr-cfp-cat-yfp*-P*spy*. Afterward, we generated a PCR product using primers MBP384/MBP385 and pKD4 as template with Q5® High-Fidelity DNA polymerase and replaced *baeR* by *aph* by recombineering[55]. The 5´ and 3´ junctions of chromosome and *aph* were afterwards verified by PCR with primers MBP386/UD2822 and MBP373/MBP387, respectively.

#### Construction of *E. coli* K-12 MG1655 *att*::P*frr-cfp-cat-yfp*-P*spy cpxR::aph*. *E. coli* K-12 MG1655 *att*::P*frr-cfp-cat-yfp*-P*spy cpxR::aph* was constructed by transforming pKD46 into *E. coli* K-12 MG1655 *att*::P*frr-cfp-cat-yfp*-P*spy*. Afterward, we generated a PCR product using the primers MBP380/MBP381 and pKD4 as a template with the Q5® High-Fidelity DNA polymerase and replaced *cpxR* with *aph* by recombineering[55]. The 5´and 3´ junctions of the chromosome and *aph* were afterwards verified by PCR with the primers MBP382/UD2822 and MBP373/MBP383, respectively.

#### Reporter module response measurements
The bacteria were grown in M9 medium supplemented with 0.4% casamino acids and 0.4% glucose in shaking conditions at 37 °C overnight in an Infors HT Multitron Standard incubator (Infors, Einsbach, Germany). The next day, the overnight cultures were inoculated 1:200 into 150 µl of freshly prepared M9 medium supplemented with 0.4% casamino acid and 0.4% glucose in black µ-clear plates with transparent bottom (Greiner Bio-One, Frickenhausen, Germany), covered with a transparent lid. Afterwards, the plate was incubated at 37 °C in a TECAN Infinite 200pro instrument (TECAN, Männedorf, Switzerland) for the indicated time with orbital shaking with an amplitude of 1.5 mm. Optical density was measured at 595 nm, and the fluorescence signal for CFP (excitation wavelength 442, emission wavelength 492, detector gain 85) and for YFP (excitation wavelength 514, emission wavelength 550, detector gain 100) were recorded automatically every 10 min. At the indicated time point, 5 µl or the

same volume of serial dilutions of the test substance, as well as a negative control (solvent) were added to the wells, and the measurement was continued until the time point of 20 h, which was the endpoint of each experiment. The optical density signal was corrected for the blank. The YFP signal was normalized to the CFP signal of each well for each time point. Promoter activity was calculated by subtracting the YFP/CFP value of the previous time point: promoter activity $[\mathrm{d(YFP/CFP)}]_n = \mathrm{(YFP/CFP)}_n - \mathrm{(YFP/CFP)}_{n-1}$. For the photosensitizing agent, after adding the chemicals to the cultures, an extra 5 min shaking was required to properly mix the agent with the bacterial cells followed by 15 min of light exposure (660 nm ± 26 nm, 10 mW/cm$^2$, 9 J/cm$^2$) to photo-activate the compounds.

#### Statistics and reproducibility
For the total response measurements, experiments were conducted in triplicate, and the results are expressed as mean ± standard deviation. Statistical data analysis was performed using GraphPad Prism 9 (GraphPad Software). Means were compared using a one-way ANOVA analysis across multiple groups. For all tests, a $P$ value < 0.05 was considered statistically significant.

#### Reporting summary
Further information on research design is available in the Nature Portfolio Reporting Summary linked to this article.

### Data availability
The source data for Figs. 3–7 are provided in Supplementary Data 1. The data that support other findings of this study are provided in the supporting information or available from the corresponding authors upon reasonable request.

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

## Acknowledgements
Our work was supported by the German Federal Ministry for Education and Research (grant no. 16GW0246) and by the Interdisciplinary Center for Clinical Research, Medical Faculty Münster (grant no. Dob2/022/16). D.T. was supported by the Medizinerkolleg (MedK) Münster program of the University of Münster. Support by the Münster Graduate School of Evolution (MGSE) to M.C. is gratefully acknowledged. A.G. thanks the German Research Foundation (GA 2362/6-1) and University of Duisburg Essen for the support. We thank Susanne Lindgren for the critical reading of the manuscript.

## Author contributions
M.B., A.G., and U.D. conceived and designed the study. M.C., A.G., and D.T. performed experiments and analyzed the data. M.C., A.G., and M.B. wrote the manuscript. All authors approved the final version of the manuscript.

## Funding

## Competing interests
The authors declare no competing interests.
