## [Transparent Peer Review file · Communications Biology]

Photosensitizer-Specific Bacterial Stress Responses Reveal Distinct Targets in Photoinduced Inactivation

Corresponding Author: Professor Anzhela Galstyan

Version 0:

Reviewer comments:

Reviewer #1

(Remarks to the Author)

Review of a manuscript submitted to Communications Biology entitled: "Photosensitizer-specific bacterial stress responses reveal distinct targets in photoinduced inactivation"

The study by Chitto et al. presents a novel approach to identifying targets for the photoinactivation of bacteria (aPDT), using *Escherichia coli* as a model organism. The authors employed two distinct photosensitizing molecules, activated by visible light and classified as Type I or Type II photosensitizers, to induce stress in bacterial cells. Furthermore, they utilized an existing yet highly sophisticated biosensor model to generate transcriptional fusions of selected gene promoters responsive to key stress factors, including pH fluctuations, osmotic stress, envelope damage, oxidative stress, and DNA damage. This approach enabled them to evaluate the activation of specific genetic elements following phototreatment. As a result, authors obtained activation of different stress-dependent pathways following aPDT depending on a photosensitizer applied.

This research is particularly significant as it underscores the importance of considering the specific photosensitizer used in photodynamic therapy (PDT) against microbes along with its light-independent mechanism of action. While most studies predominantly focus on oxidative stress—widely recognized as the principal mechanism of aPDT due to the generation of reactive oxygen species (ROS)—it is well established that additional stress-response pathways may also be activated in bacteria upon exposure to aPDT. Moreover, aPDT efficacy must be evaluated in direct relation to the specific light-activated molecule employed. Even if a photosensitizer exhibits negligible toxicity in the absence of light, this does not guarantee that it remains biologically inert. Understanding these interactions is vital for optimizing PDT strategies and broadening its potential applications in antimicrobial treatments. The presented work aligns closely with this perspective, offering significant insights into these dynamics.

It should be further emphasized that the detailed mechanism underlying photoinactivation of microorganisms remain poorly characterized. Thus, this study provides novel and valuable information on the subject.

This is an interesting investigation elucidating the molecular-level mechanisms of photoinactivation and the bacterial response to photoinactivation mediated by specific PSs.

I find this work highly intriguing, as it presents original findings and offers an explanation—albeit within the limitations of the study—of the mechanisms underlying the action of two distinct photosensitizers. At the same time, it highlights the necessity of integrating aPDT strategies with the specific photosensitizers used in research.

In this review, I point out several potential modifications that could enhance the overall quality and readability of the manuscript. The results presented will undoubtedly be of interest to researchers working in the fields of aPDT, photochemistry, photobiology, microbiology, and molecular biology.

Specific comments:

Introduction

In line 34 authors state that mutants to PDT "rarely occur", I think this should be clearly specified that none was ever isolated and characterized.

In line 55 the authors state membrane is major target of aPDT. I think this sentence is unauthorized as the aPDT process strongly depends on a photosensitizer, and photosensitizers are molecules varying significantly with their chemical and biological properties. If a photosensitizer entry the cell, passes the membrane then it does not damage the membrane but

rather operates inside the cell. It should be rather stated here that main targets for PDT depend on the PS and its localization. As there are no mechanistic analysis into the death mechanism following aPDT, I would be rather careful with such statements.

I would ask the authors to give a rationale behind choosing the two specific PSs for their studies.

Results

Figure 3 – this figure must be improved, the letters are not visible, the legends, names of axes, scales are too small, and they are hardly visible even at high magnification on the screen, please change it

The description of the experiment presented in Figure 3 (lines 121–131) is more appropriate for the Methods section. In the Results section, I would suggest focusing primarily on the findings obtained. It would be more effective to directly present the results and emphasize the main outcome in the title of the subsection.

Please specify how many times the growth curve analyses were performed and whether there were statistically significant differences between the growth curves. Additionally, I recommend including a brief 1–2 sentence summary at the end of each subchapter to consolidate the findings and enhance clarity.

The description of Figure 4 is overly detailed. The interpretation of results should be integrated into the main text, with the figure description kept concise and focused on the essential aspects of the figure.

Furthermore, please specify the duration for which the photosensitizers (PSs) were incubated with the cells prior to illumination. Was the incubation time 5 minutes? Providing this detail is essential to enhance reproducibility and ensure a clear understanding of the experimental setup.

I have no comments regarding the validation of the biosensor system, as it appears to function as intended and demonstrates some specificity.

However, the results presented in Figure 6 are not particularly convincing. Even the control experiment with H₂O₂ does not reveal a significant difference, as the variations in the behavior of the WT and mutant strains are barely distinguishable. Furthermore, how do the authors explain the observation that, in the oxyR- background, the total response is higher than in the WT, whereas under aPDT treatment, the opposite trend is observed? This apparent contradiction challenges the hypothesis regarding OxyR's involvement in MB-mediated photoinduced stress. A detailed explanation of this observation is essential to substantiate the proposed hypothesis.

Additionally, please provide detailed information on the light source used, including power density data, to ensure the reproducibility and accuracy of the experimental conditions.

The activity of stress promoters under control conditions, i.e., in the absence of an inducer (e.g., P_{gadA}, P_{otsA}, P_{dps}), requires a more detailed explanation. Furthermore, the statement that the differences are concentration-dependent seems somewhat overstated. While a trend is observable (Fig. 3 DHLPT), it should be substantiated with statistical analysis to strengthen the conclusion.

Figure S11 requires a detailed description.

In the experiments investigating the response of individual modules to MB and light (Figure S8), the control consisting of light alone is missing.

The title in line 299 should refer to aPDT rather than MB, as MB alone did not induce P_{dps} activity.

Please clarify what serves as the control in the experiment presented in Figure 6BDF. If it is H₂O, why does it induce stress promoter activity?

The authors stated (line 318) that "...P_{dps} activity was completely absent..." as shown in Figure 6C. However, the activity is present, albeit significantly reduced. Please comment on this discrepancy. If the statement is made in comparison to the control, it should be explicitly clarified to prevent misunderstanding.

Discussion

The discussion section is well-written, with the authors addressing the limitations of the model systems used in their research, such as the growth-dependent activity of reporter activation or cross-reactivity. Their efforts in this regard deserve recognition.

I did not find any statements in the discussion that extend beyond the data obtained. I have no major remarks regarding this section. However, I believe that an analysis of PS localization in *E. coli* could provide additional verification of the results. Perhaps simple fluorescence microscopy could be employed, where the signal from MB might be observed as intracellular fluorescence, while SiPc would likely yield a signal predominantly or exclusively in the cell envelope.

My general question about the study is as follows: How do the authors evaluate their approach in the context of

transcriptomic analyses? Could techniques such as RNA-seq provide additional or different insights into the mechanism of aPDT? Do the authors consider their approach to be complementary to such analyses?

Minors

I want to stress that I am not assessing the language quality, however, there were some mistakes that should be corrected, eg., double 'the' (line 302), provides constitutive expression instead provides for constitutive expression (line 367) Instead of 'more or less' (line 373) use sth like "exhibited a concentration-dependent effect, albeit to varying degrees." to be more scientifically sound.

Please add Conclusion or summary after the Discussion section

Reviewer #2

(Remarks to the Author)

This study demonstrates the potential of a novel analytical approach for discussing the intracellular targets of photosensitizers by utilizing biosensors that monitor the expression of representative genes involved in major bacterial stress response pathways. The identification of photosensitizer targets has been a long-standing issue, and this new method, by employing the SOS response, enables a comprehensive examination of major targets and clarifies whether target specificity exists. As a result, this experimental technique is expected to be highly valuable for elucidating the distinct modes of action of different photosensitizers.

In this study, the authors applied this approach and provided evidence suggesting that methylene blue (MB) and octacationic silicon phthalocyanine (SiPc) inactivate bacterial cells through different intracellular targets. However, I believe that major revisions are necessary before this manuscript can be considered for publication. Specifically, the authors should address the following points:

< Major Comments >

i) Regarding the Research Objective in the "Introduction"

As mentioned above, this study aims to facilitate the discussion of photosensitizer target sites by utilizing biosensors that monitor the expression of representative genes involved in major bacterial stress response pathways. The authors appear to place particular emphasis on applying this experimental approach to analyze the mode of action of photosensitizers. However, in the introduction, this objective remains somewhat ambiguous, and the main claims of the study should be more clearly articulated. Therefore, I suggest that the authors explicitly emphasize the central aim of this study, such as stating that this study demonstrates that this method is useful for elucidating the targets and mechanisms of antimicrobial photodynamic therapy (aPDT).

ii) Interpretation of Results

I have some concerns regarding the interpretation of the results for each photosensitizer. The authors should provide clearer explanations and ensure that their conclusions are well-supported by the data.

> Relationship Between Oxidative Stress and Photoinactivation in MB

The interpretation of the data regarding oxidative stress and photoinactivation induced by MB raises some concerns. In the presented results, at 1 μM MB, where oxidative stress was prominently detected, there was no reduction in bacterial count. At the higher concentration of 5 μM , oxidative stress levels appeared to be lower than at 1 μM , and bacterial reduction was still not observed. While it is reasonable to assume that at 1 μM , MB localizes in the cytoplasm rather than the cell membrane, these data alone do not provide sufficient evidence to conclude that oxidative stress is responsible for photoinactivation. Additionally, it remains unclear why oxidative stress was not detected at 5 μM .

Thus, it is uncertain whether cytoplasmic MB contributes to photoinactivation. In other words, the authors should discuss whether the localization and distribution of MB remain the same under conditions where photoinactivation occurs.

A related consideration is whether higher concentrations of MB are required to induce photoinactivation, or if increased light intensity or prolonged irradiation at these tested concentrations could lead to photoinactivation. In any case, the manuscript should integrate these discussions and clearly establish the logical connection between the results, interpretation, and conclusions regarding the relationship between oxidative stress and photoinactivation. Additionally, data should be added where necessary.

> Relationship Between Envelope Stress and Photoinactivation in SiPc

In the presented data, envelope stress was prominently detected at 1 μM SiPc, whereas bacterial growth was more strongly inhibited at 5 μM . This indicates a discrepancy between the concentration at which envelope stress occurs and the concentration at which bacterial growth is affected—i.e., photoinactivation is induced. Similar to the case of MB, the authors should clarify the relationship between envelope stress and photoinactivation in greater detail.

Additionally, under dark conditions at 5 μM , the final bacterial count was lower, yet under light exposure, despite a stronger impact on the growth curve, the final bacterial count was comparatively higher. This raises the question of why photoinactivation did not lead to a further reduction in bacterial count. If envelope stress occurs under dark conditions and leads to a decrease in bacterial count, it suggests that SiPc localizes to the outer membrane or cytoplasmic membrane and directly causes damage. However, if this is the case, it is also possible that light exposure inhibits this effect. One possible explanation is that SiPc undergoes photodegradation, which may interfere with the effect observed in the dark. I would be interested to hear the authors' perspective on this matter.

iii) Experimental Conditions for Light Exposure (lines 566–567)

As mentioned above, it is important to discuss the concentration of the photosensitizer and the light dose in relation to the results. If possible, I would suggest that the authors include the light intensity (e.g., W/m²) or the total energy dose (e.g., J) used during the light exposure in the "Material and Methods" (lines 566–567). This would provide a more comprehensive understanding of the experimental setup and its impact on the results.

iv) Photosensitization Reaction of MB

In lines 440–460, the authors suggest the possibility of a type I reaction being induced by MB, while in lines 467–470, they discuss the generation of singlet oxygen (type II). The authors should clarify which type of reaction (type I or type II) they believe is predominantly occurring in this study.

v) Lines 48–49

The manuscript states "several investigators"; however, only a single reference is cited. To maintain accuracy, the authors should either cite multiple studies, reference a relevant review article, or revise the wording accordingly.

vi) Lines 433–436

The authors should cite relevant literature to support the content presented in lines 433–436. This will strengthen the argument and provide a more solid foundation for the claims made in this section.

< Minor Comments >

i) Abbreviations

In lines 270, 271, 280, 299, and 345, the abbreviation "SiPC" should be corrected to "SiPc."

ii) Lines 83–84

It would be beneficial to provide additional explanations for "OxyR" as well as "BaeR and CpxR" when they are first introduced in lines 83–84. This will help readers unfamiliar with these terms to better understand their significance in the context of the study.

iii) Line 151

The terms "one hour" and "(170 min)" are unclear regarding what specific event or process they are referring to. It would be helpful to clarify and specify what these time intervals correspond to in the experimental setup.

iv) Figures

The arrows in the figure overlap with the horizontal axis of the graph. It would be better to adjust the positioning or make necessary modifications to improve clarity. Additionally, the text on the graph appears to be faint, overall. It would be advisable to make the text more prominent for better visibility.

Reviewer #3

(Remarks to the Author)

The manuscript investigates the stress responses of *Escherichia coli* K-12 MG1655 to two photosensitizers (methylene blue, MB, and silicon phthalocyanine, SiPc) in antimicrobial photodynamic therapy (aPDT). By constructing a series of stress response gene promoter-yellow fluorescent protein (YFP) reporter modules, the researchers were able to monitor the bacterial stress responses in real-time under the action of different photosensitizers. The manuscript also demonstrates that the concentration of the photosensitizer and the light activation conditions significantly influence the bacterial stress response. By deleting the relevant regulators (such as OxyR, BaeR, and CpxR), the researchers further verified the specificity of these stress responses. However, the manuscript lacks critical experimental data necessary for a comprehensive description of the damage on bacterial membrane from SiPc. Consequently, I have several concerns regarding both the content and quality of the manuscript, leading me to conclude that it cannot be accepted in its current form.

1. The author use MB and SiPc to investigate its aPDT mechanism, but there isn't any information about the purity of two molecules. The author should include H-NMR result to prove that the molecules used are pure.

2. The author are missing some bacterial membrane integrity assays to further validate the damage to the cell membrane caused by the photosensitizer, the integrity of the bacterial membrane could be assessed using a fluorescent dye such as propidium iodide (PI). This assay provides additional evidence of the impact of SiPc on membrane integrity.

3. The author should include some quantitative analysis of bacterial viability, for example by CFU counting to verify the

photodynamic antibacterial effect of the photosensitizer towards edited bacterial.

4. Although the study has revealed the types of stress responses induced by the photosensitizers, it lacks an in-depth molecular-level investigation into the mechanisms of interaction between the photosensitizers and intracellular targets within bacterial cells. For instance, it is unclear whether the oxidative stress induced by MB involves specific oxidative stress-related proteins or DNA damage repair mechanisms, and whether the envelope stress induced by SiPc is associated with particular components or structures of the cell membrane.

5. The manuscript primarily used *Escherichia coli* K-12 MG1655 as the model strain. Although this strain is representative, there may be significant differences between different bacterial species and strains. It is recommended to expand the experimental sample range in future studies, such as clinical strains with different resistance characteristics.

6. The manuscript is missing ROS results in comparing the different type of ROS producing ability between SiPc and MB.

7. In figure 5A, 5uM of SiPc have an influence on the bacterial growth rate, the manuscript is missing evidence showing that the effect on normal membrane functions.

8. The manuscript could include additional experiments, especially under light-activation conditions, to test the effects of other photosensitizers with similar structure on bacterial stress responses. This would further validate the specificity of MB and SiPc from molecular level,

Version 1:

Reviewer comments:

Reviewer #1

(Remarks to the Author)

Comments on the revised version of the manuscript by Chitto et al

The revised version of the manuscript demonstrates noticeable improvements, particularly in the quality of the figures and the clarity of some descriptions. However, certain points still require clarification to ensure accuracy and consistency. Please find below specific comments and suggestions for further revision.

Line 55 – The authors state that they have added a brief explanation regarding the selection of PSs in this study. However, my comment referred to a different issue. Specifically, I do not agree with the statement that the membrane is the primary target of PDT because there is a correlation between its damage and bacterial killing (line 55). In this case, the authors did not address my comment, and the text remains unchanged in both the previous and current versions of the manuscript. If the authors intend to refer to specific PSs (line 55), please indicate this clearly instead of using the vague expression “many PSs.”

Line 72 – Please replace “low doses” with “low concentrations” or clarify what exactly is meant by this term. Does it refer to a low concentration of the PS, or to a low dose of PDT as a whole, meaning the combined PS–light treatment?

Line 165, 174 –no. 29 should be removed

Line 194 – The title of this subchapter should be changed to: “Oxidative stress induced by light-activated MB... and envelope stress induced by light-activated SiPC...”

Line 303 – This should read: “...to the light-activated PSs to the wild type.”

The paragraph starting at line 305: I believe that the way this experiment is currently described may be problematic. The differences shown in the graphs are relatively small, which is acceptable, as this reflects the inherent variability of a biological system. For this reason, the authors included panel 6C, which summarizes the result at a specific time point. However, it is not appropriate to use the expression “activity was completely absent” when the reader can clearly see a bar on the graph (panel 6C, middle bar for *oxyR*⁻). Furthermore, in line 311, the authors state that the “peak was completely absent,” yet they show a statistically significant difference between WT and *oxyR*. Although this difference is minimal, it is misleading in the current wording. This paragraph is important and, in my opinion, should be rewritten to describe the results more accurately and clearly, for the benefit of the paper.

The quality and descriptions of the figures have been significantly improved and, in their current form, are satisfactory.

Reviewer #2

(Remarks to the Author)

Thank you for your thorough responses to my comments. I appreciate the sincere and detailed manner in which you addressed my questions and concerns. I have also reviewed the revised manuscript and confirmed that the suggested changes have been incorporated.

In particular, I now clearly understand that “the primary aim of this study was not to investigate photosensitizer (PS)-dependent bacterial killing or photoinactivation mechanisms per se, but rather to characterize the physiological stress responses of *E. coli* to sub-lethal photodynamic conditions.” I also understand that at higher concentrations, the observed signal may have been attenuated due to cellular damage. These clarifications have fully addressed my concerns.

Below are a few additional points I noticed upon reviewing the revised manuscript:

Figures

i) It appears that some of the black arrows in the figures have been replaced with dashed lines, which improves visual clarity—a positive revision. However, references to "black arrows" still remain in the main text and figure captions. I recommend updating these descriptions to reflect the changes in the figures.

ii) Additionally, in Figures 4 and 5, the original versions included black arrows at the 140-minute mark, whereas the revised figures now seem to show dashed lines near the 180-minute mark. I suggest verifying these changes to ensure consistency between the figures and their corresponding explanations.

Reviewer #3

(Remarks to the Author)

In my view, the authors have answered all the questions and revised related points, and thus this revised work can be published as it is.

Version 2:

Reviewer comments:

Reviewer #1

(Remarks to the Author)

After the authors made the changes, I think the quality of the paper is good enough to be published as is. The authors took my comments into account, and I have no further objections to publishing the analyzed results in their current form.

Reviewer #1:

Review of a manuscript submitted to Communications Biology entitled: “Photosensitizer-specific bacterial stress responses reveal distinct targets in photoinduced inactivation”

The study by Chitto et al. presents a novel approach to identifying targets for the photoinactivation of bacteria (aPDT), using *Escherichia coli* as a model organism. The authors employed two distinct photosensitizing molecules, activated by visible light and classified as Type I or Type II photosensitizers, to induce stress in bacterial cells. Furthermore, they utilized an existing yet highly sophisticated biosensor model to generate transcriptional fusions of selected gene promoters responsive to key stress factors, including pH fluctuations, osmotic stress, envelope damage, oxidative stress, and DNA damage. This approach enabled them to evaluate the activation of specific genetic elements following phototreatment. As a result, authors obtained activation of different stress-dependent pathways following aPDT depending on a photosensitizer applied.

This research is particularly significant as it underscores the importance of considering the specific photosensitizer used in photodynamic therapy (PDT) against microbes along with its light-independent mechanism of action. While most studies predominantly focus on oxidative stress—widely recognized as the principal mechanism of aPDT due to the generation of reactive oxygen species (ROS)—it is well established that additional stress-response pathways may also be activated in bacteria upon exposure to aPDT. Moreover, aPDT efficacy must be evaluated in direct relation to the specific light-activated molecule employed. Even if a photosensitizer exhibits negligible toxicity in the absence of light, this does not guarantee that it remains biologically inert. Understanding these interactions is vital for optimizing PDT strategies and broadening its potential applications in antimicrobial treatments. The presented work aligns closely with this perspective, offering significant insights into these dynamics.

It should be further emphasized that the detailed mechanism underlying photoinactivation of microorganisms remain poorly characterized. Thus, this study provides novel and valuable information on the subject.

This is an interesting investigation elucidating the molecular-level mechanisms of photoinactivation and the bacterial response to photoinactivation mediated by specific PSs. I find this work highly intriguing, as it presents original findings and offers an explanation—albeit within the limitations of the study—of the mechanisms underlying the action of two distinct photosensitizers. At the same time, it highlights the necessity of integrating aPDT strategies with the specific photosensitizers used in research.

RESPONSE: We sincerely appreciate the reviewer's thorough and insightful evaluation of the manuscript. The positive assessment of the study's novelty and significance in advancing the understanding of antimicrobial photodynamic therapy (aPDT) mechanisms is highly encouraging.

In this review, I point out several potential modifications that could enhance the overall quality and readability of the manuscript. The results presented will undoubtedly be of interest to researchers working in the fields of aPDT, photochemistry, photobiology, microbiology, and molecular biology.

RESPONSE: We would like to thank the reviewer for the thoughtful and constructive feedback and greatly appreciate the time and effort invested in providing detailed comments. It is encouraging to know that the results are considered relevant and of interest to researchers in the fields of aPDT, photochemistry, photobiology, microbiology, and molecular biology.

Specific comments:

Introduction

In line 34 authors state that mutants to PDT “rarely occur”, I think this should be clearly specified that none was ever isolated and characterized.

RESPONSE: We thank the reviewer for this valuable suggestion. The statement has been revised to clarify that, to the best of current knowledge, no resistant mutants to PDT have been isolated or characterized to date.

In line 55 the authors state membrane is major target of aPDT. I think this sentence is unauthorized as the aPDT process strongly depends on a photosensitizer, and photosensitizers are molecules varying significantly with their chemical and biological properties. If a photosensitizer entry the cell, passes the membrane then it does not damage the membrane but rather operates inside the cell. It should be rather stated here that main targets for PDT depend on the PS and its localization. As there are no mechanistic analysis into the death mechanism following aPDT, I would be rather careful with such statements. I would ask the authors to give a rationale behind choosing the two specific PSs for their studies.

RESPONSE: We thank the reviewer for this insightful and constructive comment. We agree that the mechanism of action in aPDT is highly dependent on the specific photosensitizer used, including its chemical nature, cellular uptake, and localization. We have revised the sentence accordingly to reflect that the primary targets of aPDT depend on the properties and localization of the photosensitizer, and that damage may occur at different cellular sites, not necessarily at the membrane. We also acknowledge the importance of providing a rationale for the choice of photosensitizers used in our study. We have now included a brief explanation in the revised manuscript, highlighting the distinct chemical and biological characteristics of the selected photosensitizers and their relevance to our experimental goals.

Results

Figure 3 – this figure must be improved, the letters are not visible, the legends, names of axes, scales are too small, and they are hardly visible even at high magnification on the screen, please change it

RESPONSE: We thank the reviewer for pointing this out. We agree that the readability of Figure 3 needed improvement. We have revised the figure to enhance clarity by increasing the font size of the labels, legends, axis titles, and scales. In addition, we have added an asterisk to indicate statistical significance, which was missing in the previous version of the figure. The updated version ensures that all elements are clearly visible, even at standard screen magnification. We hope the revised figure meets the reviewer’s expectations.

The description of the experiment presented in Figure 3 (lines 121–131) is more appropriate for the Methods section. In the Results section, I would suggest focusing primarily on the findings obtained. It would be more effective to directly present the results and emphasize the main outcome in the title of the subsection.

RESPONSE: Thank you for the valuable comment. Since the sensor system has not been described elsewhere, we believe that including the description of the results obtained with tests using known inducers is important to provide sufficient context for the reader. Therefore, we would prefer to retain this information in the Results section, while ensuring a clear focus on the main findings.

Please specify how many times the growth curve analyses were performed and whether there were statistically significant differences between the growth curves. Additionally, I recommend including a brief 1–2 sentence summary at the end of each subchapter to consolidate the findings and enhance clarity.

RESPONSE: We thank the reviewer for pointing this out, because this was indeed not clear. We show here a representative growth curve for different concentrations of known inducers of our sensors and for the PS (one growth curve per indicated concentration). The purpose of these growth curves was to control that the chosen concentration of test substances did not do major damage to the bacterial cells (that would have been indicated by a drop in OD or a stop in increase of OD). Hence, the growth curves merely served as control to identify concentrations of test substances that allowed for a physiological response of the cells. The latter, as well as the final optical densities at the end of the experiment we have now analyzed in detail (n= 3; Figure 3 B, D, E, F, H, J, L, N, P, R, T; 4 D, H; Figure 5 D, H; Figure 6 E, F).

The description of Figure 4 is overly detailed. The interpretation of results should be integrated into the main text, with the figure description kept concise and focused on the essential aspects of the figure.

RESPONSE: We thank the reviewer for this helpful suggestion. We have revised the description of Figure 4 to make it more concise and focused on the key elements. The interpretation of the results has been integrated into the main text to improve the flow and readability of the manuscript.

Furthermore, please specify the duration for which the photosensitizers (PSs) were incubated with the cells prior to illumination. Was the incubation time 5 minutes? Providing this detail is essential to enhance reproducibility and ensure a clear understanding of the experimental setup.

RESPONSE: We thank the reviewer for this important comment. The incubation time for the photosensitizers (PSs) prior to illumination was indeed 5 minutes. This information is provided in the Materials and Methods section of the manuscript.

I have no comments regarding the validation of the biosensor system, as it appears to function as intended and demonstrates some specificity.

RESPONSE: We thank the reviewer for their positive feedback regarding the validation of the biosensor system.

However, the results presented in Figure 6 are not particularly convincing. Even the control experiment with H₂O₂ does not reveal a significant difference, as the variations in the behavior of the WT and mutant strains are barely distinguishable. Furthermore, how do the authors explain the observation that, in the oxyR- background, the total response is higher than in the WT, whereas under aPDT treatment, the opposite trend is observed? This apparent contradiction challenges the hypothesis regarding OxyR's involvement in MB-mediated photoinduced stress. A detailed explanation of this observation is essential to substantiate the proposed hypothesis.

*RESPONSE: We believe there may have been a misunderstanding regarding the control experiment presented in Figure 6. The control condition involved treatment with the solvent used for methylene blue (MB), which in this case was **5 µl sterile water (H₂O)**, not hydrogen peroxide (H₂O₂). This was intended as a negative control to ensure that the solvent alone did not elicit any stress response. We did not observe significant differences in between wild-*

type and the **oxyR** and **mntR** mutant strains (Figure 6 F), thereby confirming that the solvent itself does not induce the *dps* promoter module.

Additionally, please provide detailed information on the light source used, including power density data, to ensure the reproducibility and accuracy of the experimental conditions.

RESPONSE: We thank the reviewer for highlighting the importance of providing detailed information on the light source. We have now included the specifications of the light source used in our experiments. The cells were irradiated with an LED lamp at 660 ± 26 nm light with a power density of 10 mW/cm^2 for 15 min. This additional detail has been added to the Methods section to ensure clarity, reproducibility, and accuracy of the experimental conditions.

The activity of stress promoters under control conditions, i.e., in the absence of an inducer (e.g., *PgadA*, *PotsA*, *Pdps*), requires a more detailed explanation.

*RESPONSE: In contrast to *Pspy* and *PrecA*, *PgadA*, *PotsA* and *Pdps* have in addition to *RpoD* (house keeping sigma factor) also *RpoS*. This sigma factor is frequently involved in the expression of stress related genes, but also of genes which are required in stationary phase. The activation of *rpoS* occurs at the transition to stationary phase, resulting in the growth phase-dependent expression of the genes in this regulon. We have now added this information in the discussion (line 416 - 417).*

Furthermore, the statement that the differences are concentration-dependent seems somewhat overstated. While a trend is observable (Fig. 3 DHLPT), it should be substantiated with statistical analysis to strengthen the conclusion.

RESPONSE: We appreciate the reviewer's comment and fully acknowledge the importance of providing statistical validation when interpreting concentration-dependent effects. We have now analysed the response of our modules to known inducers in detail ($n = 3$) and we indeed observe a statistically significant decrease in total YFP expression in response to decreasing amounts of the known inducers (Figure 3 D, H, L, P, T).

Figure S11 requires a detailed description

RESPONSE: Thank you for your feedback. We have removed Figure S11 from the supplementary materials, as it was not discussed in the main text. We agree that its inclusion without sufficient context could be confusing.

In the experiments investigating the response of individual modules to MB and light (Figure S8), the control consisting of light alone is missing.

RESPONSE: We thank the reviewer for pointing this out. The control condition consisting of light exposure without methylene blue (MB) was included in the experiment and is represented as "cont (-)" in Figure S8. In this setup, cells were treated with the solvent (water) and exposed to light, serving as the appropriate light-only control. We have clarified this more explicitly in the figure legend to avoid any confusion in the revised manuscript.

The title in line 299 should refer to aPDT rather than MB, as MB alone did not induce *Pdps* activity.

RESPONSE: We thank the reviewer for this valuable suggestion. We have revised the title in line 299 to reflect the correct focus on aPDT.

Please clarify what serves as the control in the experiment presented in Figure 6BDF. If it is H₂O, why does it induce stress promoter activity?

RESPONSE: We thank the reviewer for this important observation. In Figure 6BDF, the control condition corresponds to treatment with water, which serves as the solvent for methylene blue (MB) and was intended as a negative control. While the observed differences in reporter gene expression between the wild type and the oxyR mutant are small, they are statistically significant. However, it is important to note that OxyR functions as an activator of dps expression in response to oxidative stress. Therefore, the observed reporter activity in the oxyR mutant under control conditions cannot be attributed to oxidative stress, but rather to RpoS that might become shortly activated due to changes in growth conditions (during the treatment). A specific response to stress can only be inferred when reporter gene expression is lower in the absence of the corresponding activator, as shown in Figure 6 E, where OxyR-dependent activation becomes clearly evident under stress-inducing conditions.

The authors stated (line 318) that "...Pdps activity was completely absent..." as shown in Figure 6C. However, the activity is present, albeit significantly reduced. Please comment on this discrepancy. If the statement is made in comparison to the control, it should be explicitly clarified to prevent misunderstanding.

RESPONSE: We thank the reviewer for pointing out this discrepancy. We agree that the phrasing in line 318 was unclear. The statement has been revised to clarify that while Pdps activity was significantly reduced, it was not completely absent. We have also specified that this comparison was made relative to the control, in order to avoid any misunderstanding. The revised text ensures a more accurate and transparent presentation of the results.

Discussion

The discussion section is well-written, with the authors addressing the limitations of the model systems used in their research, such as the growth-dependent activity of reporter activation or cross-reactivity. Their efforts in this regard deserve recognition.

I did not find any statements in the discussion that extend beyond the data obtained. I have no major remarks regarding this section. However, I believe that an analysis of PS localization in *E. coli* could provide additional verification of the results. Perhaps simple fluorescence microscopy could be employed, where the signal from MB might be observed as intracellular fluorescence, while SiPc would likely yield a signal predominantly or exclusively in the cell envelope.

*RESPONSE: We thank the reviewer for the thoughtful and constructive feedback. We appreciate the recognition of our efforts in addressing the limitations of the model systems used in our study. Regarding the suggestion to analyze photosensitizer (PS) localization in *E. coli* using fluorescence microscopy, we agree that such an approach could, in principle, provide valuable complementary insights. However, due to the inherent resolution limits imposed by the diffraction of light, conventional fluorescence microscopy lacks the spatial resolution necessary to reliably distinguish between signals originating from the cytoplasm and those associated with the cell envelope in bacterial cells.*

This limitation was one of the key motivations for developing the biosensor-based approach presented in our study. By employing stress-responsive transcriptional reporters, our method enables functional localization of PS-induced stress responses

with pathway-level resolution, providing a more biologically relevant and mechanistically informative readout than fluorescence imaging alone. We believe this strategy offers a robust and scalable alternative for probing intracellular effects of photodynamic treatment in bacteria.

My general question about the study is as follows: How do the authors evaluate their approach in the context of transcriptomic analyses? Could techniques such as RNA-seq provide additional or different insights into the mechanism of aPDT? Do the authors consider their approach to be complementary to such analyses?

RESPONSE: We thank the reviewer for this insightful question. Indeed, transcriptomic approaches such as RNA-seq offer powerful, genome-wide insight into bacterial responses and could certainly provide complementary data regarding the mechanisms of antimicrobial photodynamic therapy (aPDT). However, our approach is designed to serve a distinct yet complementary purpose. The marker genes used in our system have well-characterized stress-specific promoters, whose transcriptional regulation under defined conditions has been extensively validated in previous studies and was also confirmed in our work. One of the key motivations for developing this reporter-based system—as outlined in the Introduction—was to provide a rapid, pathway-specific readout of cellular stress responses, enabling real-time monitoring with high temporal resolution. This functional insight is particularly valuable for identifying optimal concentrations and time points for more resource-intensive transcriptomic analyses. Thus, rather than replacing transcriptomic approaches, our system is intended to guide and complement them by helping to focus such analyses on biologically relevant conditions and time points after the treatment.

Minors

I want to stress that I am not assessing the language quality, however, there were some mistakes that should be corrected, eg., double 'the' (line 302), provides constitutive expression instead provides for constitutive expression (line 367) Instead of 'more or less' (line 373) use sth like "exhibited a concentration-dependent effect, albeit to varying degrees." to be more scientifically sound. Please add Conclusion or summary after the Discussion section

RESPONSE: We thank the reviewer for the helpful comments. We have addressed the suggested corrections, including the language improvements and the addition of a Conclusion section after the Discussion.

Reviewer #2:

This study demonstrates the potential of a novel analytical approach for discussing the intracellular targets of photosensitizers by utilizing biosensors that monitor the expression of representative genes involved in major bacterial stress response pathways. The identification of photosensitizer targets has been a long-standing issue, and this new method, by employing the SOS response, enables a comprehensive examination of major targets and clarifies whether target specificity exists. As a result, this experimental technique is expected to be highly valuable for elucidating the distinct modes of action of different photosensitizers. In this study, the authors applied this approach and provided evidence suggesting that methylene blue (MB) and octacationic silicon phthalocyanine (SiPc) inactivate bacterial cells through different intracellular targets. However, I believe that major revisions are necessary before this manuscript can be considered for publication. Specifically, the authors should

address the following points:

RESPONSE: We greatly appreciate the reviewer's positive assessment of the novel analytical approach presented in our study. We are pleased to hear that the potential of utilizing biosensors to monitor gene expression involved in bacterial stress response pathways is recognized as a valuable contribution to understanding the intracellular targets of photosensitizers. We are grateful for the reviewer's suggestions and have carefully address the points raised to improve the manuscript.

< Major Comments >

i) Regarding the Research Objective in the "Introduction"

As mentioned above, this study aims to facilitate the discussion of photosensitizer target sites by utilizing biosensors that monitor the expression of representative genes involved in major bacterial stress response pathways. The authors appear to place particular emphasis on applying this experimental approach to analyze the mode of action of photosensitizers. However, in the introduction, this objective remains somewhat ambiguous, and the main claims of the study should be more clearly articulated. Therefore, I suggest that the authors explicitly emphasize the central aim of this study, such as stating that this study demonstrates that this method is useful for elucidating the targets and mechanisms of antimicrobial photodynamic therapy (aPDT).

RESPONSE: We thank the reviewer for this valuable suggestion. We agree that a clearer and more direct articulation of the study's objective in the Introduction will enhance the clarity and focus of the manuscript. In response, we have revised the Introduction to explicitly state that the aim of this study is to demonstrate the utility of our biosensor-based approach in characterizing the physiological stress responses induced by antimicrobial photodynamic therapy (aPDT). The focus of the study lies in capturing pathway-specific responses to sub-lethal stress, which can provide insight into the likely cellular targets of aPDT. This distinction is now more clearly emphasized to better position our work within the broader context of aPDT research.

ii) Interpretation of Results

I have some concerns regarding the interpretation of the results for each photosensitizer. The authors should provide clearer explanations and ensure that their conclusions are well-supported by the data.

RESPONSE: We appreciate the feedback regarding the interpretation of the results for each photosensitizer. In response, we have revised the manuscript to provide clearer explanations and ensure that our conclusions are better supported by the data.

Relationship Between Oxidative Stress and Photoinactivation in MB

The interpretation of the data regarding oxidative stress and photoinactivation induced by MB raises some concerns. In the presented results, at 1 μM MB, where oxidative stress was prominently detected, there was no reduction in bacterial count. At the higher concentration of 5 μM , oxidative stress levels appeared to be lower than at 1 μM , and bacterial reduction was still not observed. While it is reasonable to assume that at 1 μM , MB localizes in the cytoplasm rather than the cell membrane, these data alone do not provide sufficient evidence to conclude that oxidative stress is responsible for photoinactivation. Additionally, it remains

unclear why oxidative stress was not detected at 5 μM .

Thus, it is uncertain whether cytoplasmic MB contributes to photoinactivation. In other words, the authors should discuss whether the localization and distribution of MB remain the same under conditions where photoinactivation occurs.

A related consideration is whether higher concentrations of MB are required to induce photoinactivation, or if increased light intensity or prolonged irradiation at these tested concentrations could lead to photoinactivation. In any case, the manuscript should integrate these discussions and clearly establish the logical connection between the results, interpretation, and conclusions regarding the relationship between oxidative stress and photoinactivation. Additionally, data should be added where necessary.

RESPONSE: We thank the reviewer for raising this important point. We would like to clarify that the primary aim of this study was not to investigate photosensitizer (PS)-dependent bacterial killing or photoinactivation mechanisms per se, but rather to characterize the physiological stress responses of E. coli to sub-lethal photodynamic conditions. As noted in the Introduction (Lines 63–64), “there is still a lack of experimental data on the physiological response of bacteria to various PSs,” and our stated goal (Lines 68–69) was to “develop high-throughput biosensors for screening the impact of PSs on microbial physiological function.” To this end, we deliberately avoided using PS concentrations or light conditions that significantly affect bacterial viability or growth, as cell death or strong growth arrest can confound or suppress reporter gene expression, limiting the interpretability of biosensor readouts. This is a key reason why, in our 96-well plate assays, we focused on sub-lethal concentrations and moderate irradiation conditions. The observed lack of photoinactivation at 1 μM and 5 μM MB, despite differential stress responses, aligns with our goal of examining pre-lethal physiological stress signaling. Notably, reporter activity was highest at 1 μM MB, suggesting an optimal stress-inducing condition without overt toxicity. In contrast, the comparatively lower response at 5 μM MB could be attributed to interference with reporter gene expression due to early or partial growth inhibition—an effect we also observed with 5 μM SiPc (Figure 5G), where OD measurements indicated a growth-arrested or growth-compromised state. This phenomenon is well-documented in other systems, such as the SOS response in E. coli, where reporter output diminishes at higher concentrations of ciprofloxacin despite increased cellular damage (Berger et al., 2019, Supplementary Figure 2). The apparent contradiction—stronger oxidative stress response at 1 μM than at 5 μM MB—may thus reflect a biological threshold beyond which stress signaling becomes uncoupled from survival, and reporter systems no longer function reliably. This highlights an important limitation of using transcriptional reporters under bactericidal conditions, and in fact, reinforces the utility of our system in identifying sub-lethal, physiologically relevant PS concentrations that precede irreversible damage. To clarify this point, we have added the following sentence to the Discussion section: “This is important, as higher concentrations of substances that kill bacteria are also very likely to interfere with the expression of reporter genes, as seen here at 5 μM MB and 5 μM SiPC (Figure 4G, Figure 5G).”

Relationship Between Envelope Stress and Photoinactivation in SiPc

In the presented data, envelope stress was prominently detected at 1 μM SiPc, whereas bacterial growth was more strongly inhibited at 5 μM . This indicates a discrepancy between the concentration at which envelope stress occurs and the concentration at which bacterial growth is affected—i.e., photoinactivation is induced. Similar to the case of MB, the authors should clarify the relationship between envelope stress and photoinactivation in greater detail.

RESPONSE: We thank the reviewer for this observation. As with MB, the apparent discrepancy between the concentration at which envelope stress is most strongly detected (1 μM SiPc) and the concentration at which bacterial growth is more substantially inhibited (5

μM SiPc) reflects a fundamental limitation of using reporter gene assays under bactericidal conditions. At higher concentrations, substances such as SiPc that induce photoinactivation can compromise cellular processes essential for transcription and translation, thereby interfering with the reporter gene system itself. As a result, the observed decrease in reporter activity at $5 \mu\text{M}$ SiPc likely reflects not a reduced stress response, but rather an overall suppression of gene expression due to impaired cellular function. This is also evident from the growth profile (Figure 5G), where $5 \mu\text{M}$ SiPc causes immediate or sustained growth inhibition, suggesting significant physiological disruption. Therefore, similar to our interpretation of the MB data, we emphasize that reporter-based measurements are most informative at sub-lethal concentrations, where stress signaling remains active and interpretable. This reinforces the utility of our biosensor platform for identifying early, pathway-specific stress responses prior to overt photoinactivation, and it highlights the importance of using physiologically relevant concentrations when probing cellular targets. We have now added this clarification to the Discussion to better contextualize the relationship between stress response and photoinactivation in the case of SiPc.

Additionally, under dark conditions at $5 \mu\text{M}$, the final bacterial count was lower, yet under light exposure, despite a stronger impact on the growth curve, the final bacterial count was comparatively higher. This raises the question of why photoinactivation did not lead to a further reduction in bacterial count. If envelope stress occurs under dark conditions and leads to a decrease in bacterial count, it suggests that SiPc localizes to the outer membrane or cytoplasmic membrane and directly causes damage. However, if this is the case, it is also possible that light exposure inhibits this effect. One possible explanation is that SiPc undergoes photodegradation, which may interfere with the effect observed in the dark. I would be interested to hear the authors' perspective on this matter.

RESPONSE: We thank the reviewer for this thoughtful and detailed observation. The seemingly paradoxical finding—that at $5 \mu\text{M}$ SiPc, bacterial growth was more strongly impaired under light exposure, yet the final colony count was higher compared to dark conditions—indeed warrants careful consideration. One likely explanation is that at $5 \mu\text{M}$, SiPc induces killing of a subpopulation of the bacterial cells, followed by the survival and outgrowth of a rest subpopulation. This phenomenon is consistent with the full growth curve data, which show delayed but eventual regrowth. Importantly, under light conditions, while growth inhibition appears more immediate, surviving cells may recover and proliferate once the photodynamic stress subsides, leading to a high final CFU count. Regarding the dark condition: although termed “dark,” the assay involved continuous optical density measurements at 595 nm, which may have led to low-level photoactivation of SiPc. This unintended light exposure could have triggered limited photodynamic effects, complicating the interpretation of “dark” controls and potentially contributing to envelope damage or cell death.

We agree that photodegradation is a factor worth considering in the interpretation of PS activity. However, silicon phthalocyanines (SiPcs) are known for their high photostability, and it is unlikely that the 15-minute irradiation applied in our experimental setup would lead to significant photodegradation. Therefore, we consider it improbable that light exposure reduces the membrane-disruptive potential of SiPc through degradation. While we did not directly assess SiPc stability in this study, the structural robustness of this compound class under comparable photodynamic conditions has been well documented, supporting the assumption that SiPc remains functionally active throughout the exposure period (Galstyan et al. Chem. Eur. J. (2016), 22(15), 5243 – 5252; Chem. Eur. J. (2018), 24, 1178-1186; Chem. Eur. J., (2021), 27(6): 1903–1920;). We appreciate the reviewer's interest in this aspect, and we have included a more nuanced discussion of these points in the revised manuscript to better reflect the complexity of PS behavior under varying conditions.

iii) Experimental Conditions for Light Exposure (lines 566–567)

As mentioned above, it is important to discuss the concentration of the photosensitizer and the light dose in relation to the results. If possible, I would suggest that the authors include the light intensity (e.g., W/m^2) or the total energy dose (e.g., J) used during the light exposure in the “Material and Methods” (lines 566–567). This would provide a more comprehensive understanding of the experimental setup and its impact on the results.

RESPONSE: We thank the reviewer for this valuable suggestion. We agree that providing more detailed information on the light exposure conditions would enhance the clarity of the experimental setup. In response, we have added the light intensity (W/m^2) and total energy dose (J) used during the light exposure to the "Materials and Methods" section.

iv) Photosensitization Reaction of MB

In lines 440–460, the authors suggest the possibility of a type I reaction being induced by MB, while in lines 467–470, they discuss the generation of singlet oxygen (type II). The authors should clarify which type of reaction (type I or type II) they believe is predominantly occurring in this study.

RESPONSE: We thank the reviewer for highlighting this important point. As discussed in the manuscript and illustrated in Figure 1, methylene blue (MB) is well known to participate in both type I (electron transfer leading to radical formation) and type II (energy transfer leading to singlet oxygen) photoreactions. Rather than favoring one exclusively, MB typically exhibits a dynamic balance between these mechanisms, with the predominant pathway depending on specific environmental factors such as oxygen concentration, substrate availability, and cellular localization. In our study, we do not claim to distinguish definitively between the two reaction types, as our experimental setup was not designed to dissect this mechanistic detail. However, we have clarified in the revised text that both type I and type II mechanisms may contribute to the observed stress responses, and that their relative contribution likely varies depending on cellular context and local microenvironments during photodynamic exposure.

v) Lines 48–49

The manuscript states "several investigators"; however, only a single reference is cited. To maintain accuracy, the authors should either cite multiple studies, reference a relevant review article, or revise the wording accordingly.

RESPONSE: We thank the reviewer for pointing this out. Indeed, only a single reference was cited in this sentence. To maintain accuracy, we have revised the wording to reflect that this conclusion is based on the findings of that particular study, rather than implying consensus among multiple investigators.

vi) Lines 433–436

The authors should cite relevant literature to support the content presented in lines 433–436. This will strengthen the argument and provide a more solid foundation for the claims made in this section.

RESPONSE: We thank the reviewer for this helpful suggestion. In addition to citing the relevant literature with respect to the regulation of the modules to support the statements

made in lines 433–436, we have also included live/dead CFU counting data in the Supporting Information to provide further experimental support and strengthen the conclusions drawn in this section.

< Minor Comments >

i) Abbreviations

In lines 270, 271, 280, 299, and 345, the abbreviation "SiPC" should be corrected to "SiPc."

RESPONSE: Done.

ii) Lines 83–84

It would be beneficial to provide additional explanations for "OxyR" as well as "BaeR and CpxR" when they are first introduced in lines 83–84. This will help readers unfamiliar with these terms to better understand their significance in the context of the study.

RESPONSE: Done

iii) Line 151

The terms "one hour" and "(170 min)" are unclear regarding what specific event or process they are referring to. It would be helpful to clarify and specify what these time intervals correspond to in the experimental setup.

RESPONSE: We thank the reviewer for this observation. We have clarified the wording in the manuscript to specify that the time point "170 min" refers to the moment of acetic acid addition. The phrase "one hour" denotes the period immediately following this addition, during which the cumulative YFP/CFP reporter signal was quantified. This interval was chosen to isolate the specific response to acetic acid before the growth phase-dependent activation of P_{gdaA} began to dominate.

iv) Figures

The arrows in the figure overlap with the horizontal axis of the graph. It would be better to adjust the positioning or make necessary modifications to improve clarity. Additionally, the text on the graph appears to be faint, overall. It would be advisable to make the text more prominent for better visibility.

RESPONSE: We thank the reviewer for these suggestions we corrected that and hope that this has improved the clarity.

Reviewer #3

The manuscript investigates the stress responses of *Escherichia coli* K-12 MG1655 to two photosensitizers (methylene blue, MB, and silicon phthalocyanine, SiPc) in antimicrobial photodynamic therapy (aPDT). By constructing a series of stress response gene promoter-yellow fluorescent protein (YFP) reporter modules, the researchers were able to monitor the bacterial stress responses in real-time under the action of different photosensitizers. The manuscript also demonstrates that the concentration of the photosensitizer and the light activation conditions significantly influence the bacterial stress response. By deleting the relevant regulators (such as OxyR, BaeR, and CpxR), the researchers further verified the specificity of these stress responses. However, the manuscript lacks critical experimental data necessary for a comprehensive description of the damage on bacterial membrane from SiPc. Consequently, I have several concerns regarding both the content and quality of the manuscript, leading me to conclude that it cannot be accepted in its current form.

RESPONSE: We greatly appreciate the reviewer's constructive feedback, which will help us improve the quality and completeness of the manuscript.

1. The author use MB and SiPc to investigate its aPDT mechanism, but there isn't any information about the purity of two molecules. The author should include H-NMR result to prove that the molecules used are pure.

RESPONSE: We thank the reviewer for raising this point. Regarding methylene blue (MB), we would like to clarify that it was commercially available, and as such, its purity is assumed to meet standard specifications from the supplier. For silicon phthalocyanine (SiPc), we have already published analytical data, including purity analysis, in a previous manuscript, which has been cited in the current work.

2. The author are missing some bacterial membrane integrity assays to further validate the damage to the cell membrane caused by the photosensitizer, the integrity of the bacterial membrane could be assessed using a fluorescent dye such as propidium iodide (PI). This assay provides additional evidence of the impact of SiPc on membrane integrity.

RESPONSE: We thank the reviewer for this valuable suggestion. While membrane integrity assays such as propidium iodide (PI) staining can indeed provide important insights into membrane damage, this type of analysis lies beyond the scope of our current study. As stated in the manuscript, our focus was not on elucidating killing mechanisms per se, but rather on identifying the primary physiological sites of stress response activation upon sub-lethal exposure to photosensitizers. Our biosensor-based approach was specifically designed to detect early, pathway-specific cellular responses—prior to irreversible damage or cell death.

3. The author should include some quantitative analysis of bacterial viability, for example by CFU counting to verify the photodynamic antibacterial effect of the photosensitizer towards edited bacterial.

RESPONSE: We thank the reviewer for this important suggestion. In response, we have now included quantitative analysis of bacterial viability by CFU counting to verify the photodynamic antibacterial effect of the photosensitizers. These data have been added to the Supporting Information.

4. Although the study has revealed the types of stress responses induced by the photosensitizers, it lacks an in-depth molecular-level investigation into the mechanisms of interaction between the photosensitizers and intracellular targets within bacterial cells. For

instance, it is unclear whether the oxidative stress induced by MB involves specific oxidative stress-related proteins or DNA damage repair mechanisms, and whether the envelope stress induced by SiPc is associated with particular components or structures of the cell membrane.

RESPONSE: We thank the reviewer for their thoughtful feedback. The primary aim of this study was to establish a straightforward and high-throughput method for detecting pathway-specific stress responses induced by photosensitizers using biosensor strains. Our data demonstrate that oxidative stress induced by MB activates the OxyR regulon, and that envelope stress induced by SiPc activates the BaeR pathway. Both OxyR and BaeR are well-established transcriptional regulators of stress response genes, and therefore directly linked to the expression of stress-related proteins. These findings validate the utility of our biosensor system for identifying the primary physiological responses to different photosensitizers and can serve as a basis for future studies exploring specific molecular interactions in greater depth.

5. The manuscript primarily used Escherichia coli K-12 MG1655 as the model strain. Although this strain is representative, there may be significant differences between different bacterial species and strains. It is recommended to expand the experimental sample range in future studies, such as clinical strains with different resistance characteristics.

RESPONSE: We thank the reviewer for this valuable suggestion. In this study, we chose E. coli K-12 MG1655 as a well-characterized and widely used model strain, which allowed for controlled and reproducible experiments. We do not expect significant deviations from the core mechanistic insights presented here when using other strains. And believe that using a single, standardized strain provides a solid foundation for our study.

6. The manuscript is missing ROS results in comparing the different type of ROS producing ability between SiPc and MB.

RESPONSE: We thank the reviewer for this valuable feedback. Methylene blue (MB) is a well-known standard photosensitizer, and its ROS-producing ability is well-established in the literature. As for silicon phthalocyanine (SiPc), the ROS production data have been previously published and cited in our manuscript.

7. In figure 5A, 5uM of SiPc have an influence on the bacterial growth rate, the manuscript is missing evidence showing that the effect on normal membrane functions.

RESPONSE: We thank the reviewer for this observation. Our data in Figure 5A demonstrate that 5 μ M SiPc influences bacterial growth, and we further show that it induces envelope stress responses. While we did not explicitly claim that SiPc disrupts normal membrane function, the observed envelope stress suggests that membrane-related processes are indeed affected and eventually completely impaired at higher SiPc concentrations.

8. The manuscript could include additional experiments, especially under light-activation conditions, to test the effects of other photosensitizers with similar structure on bacterial stress responses. This would further validate the specificity of MB and SiPc from molecular level,

RESPONSE: We thank the reviewer for the constructive suggestion. We agree that additional experiments under light-activation conditions, testing other photosensitizers with similar structures, could provide valuable insights and further validate the specificity of MB and SiPc at the molecular level. We will certainly consider this approach in our future studies to expand on the current findings and provide a broader understanding of the effects of different photosensitizers on bacterial stress responses.

Reviewer #1 (Remarks to the Author):

Comments on the revised version of the manuscript by Chitto et al

The revised version of the manuscript demonstrates noticeable improvements, particularly in the quality of the figures and the clarity of some descriptions. However, certain points still require clarification to ensure accuracy and consistency. Please find below specific comments and suggestions for further revision.

RESPONSE: We are glad to hear that the revisions have led to noticeable improvements, especially in the figures and clarity of the descriptions. We appreciate your detailed comments and suggestions, and we will address each point carefully to ensure the manuscript achieves greater accuracy and consistency.

Line 55 – The authors state that they have added a brief explanation regarding the selection of PSs in this study. However, my comment referred to a different issue. Specifically, I do not agree with the statement that the membrane is the primary target of PDT because there is a correlation between its damage and bacterial killing (line 55). In this case, the authors did not address my comment, and the text remains unchanged in both the previous and current versions of the manuscript. If the authors intend to refer to specific PSs (line 55), please indicate this clearly instead of using the vague expression “many PSs.”

RESPONSE: We thank the reviewer for the clarification. The text has been revised replacing the vague expression “many PSs.” We have modified the statement to avoid the implication that membrane damage is the sole or primary mechanism of aPDT. We acknowledge that while membrane damage can correlate with bacterial killing in some cases, aPDT involves multiple targets and mechanisms. To support this clarification, we have added new references in the revised manuscript.

Line 72 – Please replace “low doses” with “low concentrations” or clarify what exactly is meant by this term. Does it refer to a low concentration of the PS, or to a low dose of PDT as a whole, meaning the combined PS–light treatment?

RESPONSE: We appreciate the reviewer’s suggestion. In the revised manuscript, we have replaced the term “low doses” with “low concentrations” to more accurately reflect that we are referring to the concentration of the photosensitizer.

Line 165, 174 –no. 29 should be removed

Line 194– The title of this subchapter should be changed to: “Oxidative stress induced by light-activated MB... and envelope stress induced by light-activated SiPC...”

Line 303 – This should read: “...to the light-activated PSs to the wild type.”

RESPONSE: Done.

The paragraph starting at line 305: I believe that the way this experiment is currently described may be problematic. The differences shown in the graphs are relatively small, which is acceptable, as this reflects the inherent variability of a biological system. For this reason, the authors included panel 6C, which summarizes the result at a specific time point. However, it is not appropriate to use the expression “activity was completely absent” when the reader can clearly see a bar on the graph (panel 6C, middle bar for oxyR⁻).

RESPONSE: We thank the reviewer for this comment. The actual sentence was “In the mntR-negative strain background, the response of the Pdps module to light activation of 1 μM MB was very similar to wild type and showed a prominent peak (Figure 6 A, 150 min) in Pdps module activity directly following the treatment (Figure 6 A, dashed line), that was completely absent in the control experiment (Figure 6 B).”, i.e. we were referring here to the

absence of a prominent peak in promoter activity and not a an absence of overall promoter activity. We are sorry, if this was not sufficiently clear. We have now rephrased the section.

In the *mntR*-negative strain background, the response of the Pdps module to light activation of 1 μ M MB was very similar to wild type and showed a prominent peak (Figure 6 A, 150 min) in Pdps module activity directly following the treatment (Figure 6 A, black arrow). This peak in promoter activity was completely absent in the control experiment (Figure 6 B). Notably, this peak in Pdps activity following the treatment was as well completely absent in the *oxyR* mutant (Figure 6 C) and the module showed a very similar response in comparison to the water control experiment in the *oxyR* mutant (Figure 6 D).

We hope the section is now sufficiently clear.

Furthermore, in line 311, the authors state that the “peak was completely absent,” yet they show a statistically significant difference between WT and *oxyR*. Although this difference is minimal, it is misleading in the current wording. This paragraph is important and, in my opinion, should be rewritten to describe the results more accurately and clearly, for the benefit of the paper.

RESPONSE: The statistically significant difference in between WT and oxyR in Figure 6 C is due to the absence of induction of the dps promoter in the mutant, which we hope is now clear (missing peak of activity; see above). That the basal dps promoter activity in between wild type and oxyR regulator mutant is statistically different (Figure 6 F) not surprising. However, we would like to not that the basal promoter activity under control conditions is slightly higher in the oxyR mutant than in the wild type as opposed to the treatment (Figure 6 E), further emphasizing the importance of OxyR for MB dependent activation of the promoter.

The quality and descriptions of the figures have been significantly improved and, in their current form, are satisfactory.

RESPONSE: We sincerely thank the reviewer for the positive feedback. We are pleased to hear that the quality and descriptions of the figures have been significantly improved and are now considered satisfactory.

Reviewer #2 (Remarks to the Author):

Thank you for your thorough responses to my comments. I appreciate the sincere and detailed manner in which you addressed my questions and concerns. I have also reviewed the revised manuscript and confirmed that the suggested changes have been incorporated.

In particular, I now clearly understand that “the primary aim of this study was not to investigate photosensitizer (PS)-dependent bacterial killing or photoinactivation mechanisms per se, but rather to characterize the physiological stress responses of *E. coli* to sub-lethal photodynamic conditions.” I also understand that at higher concentrations, the observed signal may have been attenuated due to cellular damage. These clarifications have fully addressed my concerns.

RESPONSE: Thank you very much for your kind and thoughtful feedback. We are grateful that our detailed responses and revisions have satisfactorily addressed your questions and concerns.

Below are a few additional points I noticed upon reviewing the revised manuscript:

Figures

i) It appears that some of the black arrows in the figures have been replaced with dashed lines, which improves visual clarity—a positive revision. However, references to "black arrows" still remain in the main text and figure captions. I recommend updating these descriptions to reflect the changes in the figures.

RESPONSE: Thank you for highlighting this detail. We appreciate your positive note regarding the improvement in visual clarity from replacing some black arrows with dashed lines. We have carefully updated the main text and figure captions to accurately reflect these changes, ensuring consistency between the descriptions and the figures.

ii) Additionally, in Figures 4 and 5, the original versions included black arrows at the 140-minute mark, whereas the revised figures now seem to show dashed lines near the 180-minute mark. I suggest verifying these changes to ensure consistency between the figures and their corresponding explanations.

RESPONSE: We thank the reviewer for carefully checking the figures. Figures have been corrected and the dashed lines added at the proper induction time.

Reviewer #3 (Remarks to the Author):

In my view, the authors have answered all the questions and revised related points, and thus this revised work can be published as it is.

RESPONSE: We are pleased to know that all questions and related points have been satisfactorily addressed, and we truly appreciate your recommendation that the revised manuscript is suitable for publication in its current form.